# Characterizing source fingerprints and ageing processes in laboratory-generated secondary organic aerosols using proton-nuclear magnetic resonance ($^1$H-NMR) analysis and HPLC HULIS determination.

Zanca, N.[1,5], A. T. Lambe[2,3], P. Massoli[2], M. Paglione[1], D. R. Croasdale[3], Y. Parmar[3], E. Tagliavini[4], S. Gilardoni[1], S. Decesari[1]

[1]Institute of Atmospheric Sciences and Climate (ISAC) of the National Research Council of Italy (CNR), Bologna, 40129, Italy.

[2]Aerodyne Research Inc., Billerica, MA 01821, USA.

[3]Chemistry Department, Boston College, Chestnut Hill, MA 02467, USA.

[4]Department of Chemistry "Giacomo Ciamician", University of Bologna, Bologna, 40126, Italy.

[5]Proambiente S.c.r.l., Bologna, 40129, Italy.

*Corresponding author: Nicola Zanca (*n.zanca@consorzioproambiente.it)

**Abstract.** The study of secondary organic aerosol (SOA) in laboratory settings has greatly increased our knowledge of the diverse chemical processes and environmental conditions responsible for the formation of particulate matter starting from biogenic and anthropogenic volatile compounds. However, characteristics of the different experimental setups and the way they impact the composition and the timescale of formation of SOA are still subject to debate. In this study, SOA samples were generated using a Potential Aerosol Mass (PAM) oxidation flow reactor using α-pinene, naphthalene and isoprene as precursors. The PAM reactor facilitated exploration of SOA composition over atmospherically-relevant photochemical aging time scales that are unattainable in environmental chambers. The SOA samples were analyzed using two state-of-the-art analytical techniques for SOA characterization - proton nuclear magnetic resonance ($^1$H-NMR) spectroscopy and HPLC determination of humic-like substances (HULIS). Results were compared with previous Aerodyne aerosol mass spectrometer (AMS) measurements. The combined $^1$H-NMR, HPLC, and AMS datasets show that the composition of the studied SOA systems tend to converge to highly oxidized organic compounds upon prolonged OH exposures. Further, our $^1$H-NMR findings show that only α-pinene SOA acquire spectroscopic features comparable to those of ambient OA when exposed to at least $1*10^{12}$ molec OH /cm$^3$*s OH exposure, or multiple days of equivalent atmospheric OH oxidation. Over multiple days of equivalent OH exposure, the formation of HULIS is observed in both α-pinene SOA and in naphthalene SOA (maximum yields: 16% and 30%, respectively, of total analyzed WSOC), providing evidence of the formation of humic-like polycarboxylic acids in unseeded SOA.

## 1 Introduction

Organic aerosol (OA) constitutes a large proportion of ambient particulate matter, affecting the Earth's radiation balance, cloud formation and human health (Hallquist et al., 2009). Understanding and simulating the concentration and composition of OA particles is one of the major challenges of modern atmospheric chemistry. In the mid 2000's the discovery that oxidized organic compounds dominate in concentration

compared to that of primary organic compounds outside urban areas (Zhang et al., 2007; Jimenez et al., 2009), together with the understanding that the ambient organic aerosol concentrations were systematically under-predicted by existing chemical transport models (Heald et al., 2005), led to a reevaluation of the treatment of secondary organic aerosol (SOA) formation processes in chemistry and climate models. Since the model-measurement gap is mostly overcome by subjecting the particles to "oxidative aging", understanding the

nature of ageing processes has become a primary objective of new generation SOA studies. Experimental findings showing the existence of highly-oxidized SOA molecular tracers with a high oxygen-to-carbon (O/C) ratio (Szmigielski et al., 2007) and  molecular structures that are chemically distinct from 1[st] and 2[nd] generation oxidation products of the same precursors (Jenkin et al., 2000) provided indirect confirmation  of still unknown chemical processes forming highly oxidized, low-volatility compounds.

The first formulations of SOA ageing into models were based on the chemistry of saturated hydrocarbon oxidation by OH, for which a step-by-step process with a slow, progressive increase of the oxidation state, along with a decrease in volatility, can be proposed (Robinson et al., 2007). The duration of such processes clearly exceeds the residence time of SOA in traditional environmental chamber experiments with equivalent atmospheric ageing times of less than 1 day. These limitations led to the emergence of oxidation flow reactors

that are capable of higher integrated oxidant exposures, including the Potential Aerosol Mass (PAM) oxidation flow reactor (Kang et al., 2007; Lambe et al., 2011) and related techniques (Hall IV et al., 2013; Keller and Burtscher, 2012; Slowik et al., 2012). Recent studies suggest that flow reactor-generated SOA particles have similar composition to SOA generated in chambers (Lambe et al., 2015; Bruns et al., 2015). Modeling work further suggests that flow reactors simulate tropospheric oxidation reactions with minimal

experimental artifacts (Li et al., 2015; Peng et al., 2015, 2016). Recent applications of oxidation flow reactors in field measurements showed that the maximum yields of SOA were attained at approximately 2-3 days of equivalent atmospheric OH oxidation; at higher photochemical age, SOA yields decrease substantially (Tkacik et al., 2014; Ortega et al., 2016; Palm et al., 2016). Such observations demonstrate the influence of fragmentation reactions in which oxidation leads to C-C bond cleavage with the production of highly volatile

products (Kroll et al., 2009; Chacon-Madrid and Donahue 2011; Lambe et al., 2012).

The idea of a slow, multi-generation SOA ageing was recently challenged by recent findings from reaction chamber experiments employing modern chemical ionization mass spectrometric methods. For example, it was found that oxidized gaseous compounds with O/C > 0.7 form readily upon VOC oxidation (Ehn et al., 2012, 2014; Krechmer et al., 2015; Rissanen et al., 2014) and that even the chemical tracers of "aged" SOA

can be in fact produced among 2[nd] generation oxidation products (Müller et al., 2012). The quantification of highly oxidized SOA compounds in reaction chambers is challenging because of significant vapor and particle wall losses (Matsunaga and Ziemann, 2010; Zhang et al., 2014; Krechmer et al., 2016; Ye et al., 2016), but these findings suggest that SOA ageing can be much faster than previously thought (Hodzic et al., 2016). As a result of the diverse implementations of SOA schemes in models, the quantification of SOA

production and concentration in the atmosphere is still highly uncertain: a recent intercomparison between 20 state-of-the-art global models showed that the estimated SOA annual production rates differ of one order of magnitude (Tsigaridis et al., 2014). These results call for more experimental observations for constraining the existing SOA parameterizations.

The present study focuses on laboratory production of SOA from three different precursors using a PAM

reactor. The novel feature of this work is our application of two off-line analytical techniques that provide valuable insight in regards to SOA composition yet are rarely employed for SOA characterization. The first technique, [1]H-NMR spectroscopy, is a universal technique in organic chemistry. It was used to confirm the molecular structures of many SOA tracers (Finessi et al., 2014) or for following SOA reaction products in aqueous solution (Yu et al., 2011). The few examples of [1]H-NMR spectroscopy on SOA complex mixtures

(Cavalli et al., 2006; Baltensperger et al, 2008; Bones et al., 2010; White et al. 2014) indicate that the technique can be very specific for distinguishing different biogenic and anthropogenic SOA systems. The present study is among the first applications of [1]H-NMR spectroscopy to SOA samples produced from the OH oxidation of biogenic and anthropogenic SOA in the laboratory. The acquisition of NMR fingerprints for fresh and aged biogenic and anthropogenic SOA can be useful also for interpreting factor analysis results

obtained on a timeline of NMR spectra of ambient aerosol extracts (e.g., Paglione et al. 2014a). The second technique is a HPLC set-up for the determination of humic-like substances (HULIS). HULIS have been observed in ambient organic aerosol for nearly two decades (Havers et al., 1998; Limbeck et al., 2003), but their formation pathways aside from production in biomass burning plumes remain unclear (Graber and Rudich 2006). It is well-known that high-molecular weight oxygenated organic compounds readily form by

heterogeneous reactions (Limbeck et al., 2003) or by gas-to-particle conversion (Kalberer et al., 2006), but there is little evidence for their identification with HULIS in ambient aerosols (especially if we base the definition on the chromatographic behavior, as in Baduel et al., 2009).

Here we focus on SOA systems generated from three distinct precursors: isoprene, α-pinene and naphthalene. Naphthalene is used as proxy for anthropogenic aromatic intermediate volatility organic compounds

(IVOCs). Alpha-pinene is the most studied biogenic monoterpene due to its global importance as a biogenic SOA precursor (e.g. Pye et al., 2010), while isoprene is the most abundant biogenic VOC, accounting for 44% of global emissions (Guenter et al., 1995). The discovery of isoprene SOA is relatively recent (Claeys et al. 2004). In the presence of acidic wet aerosols, SOA originates from the heterogeneous uptake of isoprene epoxides ("IEPOX channel", Lin et al., 2012). Since aerosol water and acidity are primarily determined by

anthropogenic mineral acids, the formation of SOA from isoprene appears to be very much controlled by anthropogenic emissions. On the other hand, recent experiments conducted at very low nitrogen oxide (NO) concentrations, and in the absence of seed aerosols, showed that SOA can still form from isoprene ("non-IEPOX" SOA, Krechmer et al., 2015). Such aerosols are more representative of the preindustrial world and their characterization is of paramount importance for understanding the climate radiative forcing of SOA at

the global scale. Our results, obtained in the PAM reactors in absence of NOx, are representative for non-IEPOX isoprene SOA.

## 2 Experimental methods

### 2.1 PAM oxidation flow reactor

The PAM oxidation flow reactor is a horizontal 13L glass cylindrical chamber that is 46 cm long x 22 cm ID. Carrier gas flows of 8.5 liter per minute (lpm) $N_2$ and 0.5 lpm $O_2$ were used, with 8.5 lpm of flow pulled through the reactor and 0.5 lpm of excess flow removed prior to the reactor. Other experimental details are fully described in Lambe et al. (2011). In this study, the PAM reactor was connected to a Scanning Mobility Particle Sizer (SMPS), an Aerodyne time-of-flight aerosol mass spectrometer (AMS), and a filter holder

equipped with 47 mm (prebaked) quartz-fiber filters. SOA concentrations calculated from SMPS and/or AMS measurements averaged over filter collection times provided an estimate of the organic matter loading on the filters.

During the first set of experiments involving α-pinene and naphthalene as SOA precursors, by varying the concentrations of OH inside the PAM reactor, SOA with different oxidation state could be obtained. For

instance, the OH exposure varied from $2.0x10^{11}$ molec./cm$^3$*s to $2.1x10^{12}$ molec./cm$^3$*s between the α-pinene experiments and the resulting SOA oxidation degree – traced by the "f44" parameter (i.e., the fraction of the m/z 44 signal with respect to the total OA) – increased from 0.05 to 0.24. A total of five α-pinene and five naphthalene SOA samples were obtained (collection time between 3 and 20 h), with integrated OH exposures varying between $2x10^{11}$ and $2x10^{12}$ molecules/cm$^3$*s, corresponding to a photochemical age of 1.5

to 15 days assuming a 24-hour average OH concentration of $1.5*10^6$ molec cm$^{-3}$ (Mao et al., 2009).

During the second set of experiments, isoprene SOA samples were generated in the reactor (Table 2). Due to the lower yields of SOA produced by isoprene oxidation, samples were collected at OH exposure of approximately $8*10^{11}$ molec cm$^{-3}$ sec  (corresponding to a photochemical age of 6 days) at which the maximum SOA yield is obtained (Lambe et al., 2015). The collection time was varied between 2 and 18 h.

To compare SOA formation processes occurring in oxidation flow reactors and in the atmosphere, two primary assumptions are required. First, we assume the kinetics of laboratory processes occurring at higher oxidant concentrations and shorter exposure times can be extrapolated to atmospheric processes occurring at lower oxidant concentrations and longer residence times. Second, we assume that the extent of nucleation or phase partitioning of SOA is not limited by the shorter residence time in flow reactors. The first assumption
is supported by Renbaum and Smith (2011), Bahreini et al. (2012), and Lambe et al. (2015). The second assumption may introduce uncertainty depending on the particle surface area available to promote condensation and the mass accommodation coefficient of the oxidized vapors (Lambe et al., 2015; Shantanu et al., 2017).

## 2.2 Extraction and off-line sample characterization

Each filter was extracted with 5 mL of deionized ultra-pure water (Milli-Q) in a mechanical shaker for 1 h and the water extract was filtered on PTFE membranes (pore size: 0.45 μm) in order to remove suspended particles. The water extracts were dried by rotary evaporator and were then re-dissolved in 2.15mL of $D_2O$:
0.65mL for proton-nuclear magnetic resonance ($^1$H-NMR) characterization (Decesari et al., 2000), and 1.5mL for HPLC analysis and total organic carbon (TOC) analysis (Mancinelli et al., 2007). Tests of extraction using methanol instead of water were carried out on three isoprene SOA samples. The $^1$H-NMR spectra of methanol extracts were completely consistent with those obtained for the other three analyzed in deuterated water (Fig. S2), indicating that there were no specific classes of water-insoluble compounds in the
isoprene SOA under the conditions used in this study. The following discussion will focus on the water-soluble fraction for which spectroscopic and chromatographic data were obtained for all three SOA systems.

## 2.3 NMR spectroscopy

The $^1$H-NMR spectra were acquired at 600MHz with a Varian 600 spectrometer in a 5mm probe with 0.65mL
of each sample re-dessolved in $D_2O$. Sodium 3-trimethylsilyl-(2,2,3,3-d4) propionate (TSP-$d_4$) was used as the referred internal standard. A buffer of potassium deuterated formate/formic acid (pH~3.8) was used in the second series of experiments (isoprene SOA) to stabilize the chemical shift of hydrogen atoms in acyl functional groups, while the extracts obtained during the first experiments (α-pinene and naphthalene) were analyzed unbuffered. $^1$H-NMR spectroscopy of low-concentration samples in protic solvents provides the
speciation of hydrogen atoms bound to carbon atoms (H-C). On the basis of the range of frequency shifts (the chemical shift, in ppm) in which the NMR resonances occur, they can be attributed to different H-C containing functional groups (Paglione et al. 2014a).

## 2.4 HPLC-UV-TOC method

HULIS were determined using the ion exchange chromatographic method described by Mancinelli et al. (2007). A HPLC system (Agilent Model 1100) with gradient elution was used. The subsequent elution of chemical compounds bearing zero, one, two or more than two ionized groups per molecule (mainly carboxylate ions at pH 7) is monitored by an UV detector at 260 nm. Downstream of the detector, a fraction collector is programmed to sample separately "neutral compounds" (NC), "monocarboxylic acids" (MA),
"dicarboxylic acids" (DA), and "polycarboxylic acids" (PA) or HULIS. The amount of WSOC recovered in each fraction is determined off-line by TOC analysis using an Analytik-Jena multi analyzer N/C (model 2100). The HPLC column and chromatographic conditions used in this study were the same as in Mancinelli et al. (2007). Further information on the nature of the chemical classes separated by the HPLC method based on elution tests of standard compounds, including discussion of possible misclassification, is reported by
Decesari et al. (2005).

### 3.1 Results: [1]H-NMR results

### 3.1.1 NMR fingerprints of fresh and aged α-pinene SOA

Figure 1 shows the [1]H-NMR spectra of α-pinene SOA with increasing photochemical age. The first spectrum corresponding to a "low" SOA oxidation level is similar to reported NMR spectra of environmental chamber-generated α-pinene ozonolysis (Cavalli et al., 2006). However, the NMR fingerprint of α-pinene SOA evolves rapidly with further oxidation steps. A clear, progressive disappearing of 1[st] generation oxidation products (pinic and pinonic acid) with increasing O/C ratio can be observed. In the NMR spectra corresponding to a "medium" SOA oxidation level, the resonance at 0.83 ppm of chemical shift, arising from one of the two gem-methyls of pinonic acid, accounts for only 0.3% of the total integral of the spectrum, while it represented 2 – 3 % in the fresh SOA samples. This indicates that α-pinene SOA composition evolves rapidly towards more highly-oxidized molecular structures with little resemblance to  1[st] generation oxidation products. At "medium" and "high" oxidation levels, the unsubstituted alkyl groups of the SOA mixture give rise to a broad Gaussian band between 1.1 and 1.8 ppm of chemical shift with a maximum at 1.4 – 1.5 ppm. The middle point position showing a slight deshielding with respect to a purely alkylic chain (∼ 1.3 ppm for fatty acids) indicates the presence of electronegative groups (such as oxygen atoms) in beta or gamma position with respect to these alkyl groups. The band between 1.9 and 3 ppm, attributable to C-H groups of acyl groups (HC-C=O), also shows a transition towards structures containing more deshielded H atoms. In all cases, the most conspicuous band in this spectral region is found at 2.2 – 2.3 ppm of chemical shift, which corresponds to acetyl and acyl groups of aliphatic compounds with a low O/C ratio, like pinonic acid (R-$CH_2$-C=O and $CH_3$-(C=O)-R, where R is mainly a $C_xH_{2x+1}$ radical). Such band persists in all SOA samples, but an additional band between 2.5 and 2.9 ppm is observed in the two samples with the highest O/C ratio, indicating that the aliphatic groups become more and more substituted by electronegative groups: the keto and carboxylic groups become spaced by no more than two methylene (or methyns) groups (x(C=O)-CH-CH-(C=O)X, where X is a generic substituent). Finally, the NMR resonances in the third important aliphatic region of alcoxyl groups (CH-O), between 3.3 and 4.2 ppm of chemical shift, are always relatively small, with peak intensity at intermediate photochemical age. SOA species that contribute to NMR resonance in this region may be correlated with semivolatile, highly functionalized species that contribute to maximum SOA yields observed at intermediate OH exposures (Lambe et al., 2015).

Overall, the NMR fingerprint of α-pinene SOA is highly dependent on photochemical age, with a sharp change already at medium ageing. The most oxidized samples show spectral features that have lost any clear similarity with those of SOA sampled in reaction chambers experiments without ageing (Cavalli et al., 2006).

### 3.1.2 NMR fingerprints of fresh and aged naphthalene SOA

The [1]HNMR spectra of naphthalene SOA samples with increasing O/C ratio are presented in Figure 2. The extract of the least-oxidized sample (on the top) shows broad resonances between 0 and 3 ppm probably due to the effect of colloidal hydrophobic material in solution. Despite such artifact, all spectra of fresh and moderately aged SOA show clear signals from aromatic structures (region at chemical shift from 6.5 to 8.5 ppm) and alkenes (approximately between 5 and 7 ppm). The HNMR spectra of naphthalene SOA are very different than HNMR spectra of SOA produced from the OH oxidation of one-ring aromatic VOCs (Baltensperger et al., 2008), which have mostly aliphatic groups originating from ring opening reactions. Our data are in agreement with molecular speciation studies (Lee and Lane 2009), indicating that naphathalene is oxidized to form 1- or 2-ring aromatic compounds such as naphthol, and substituted benzoquinones, cinnamic acid, and phthalic acid (Chhabra et al., 2015). The chemical shift range of the main aromatic band, between 7.4 and 8.1 ppm, indicates that aromatic rings are substituted prevalently by electron-withdrawing groups, such as carbonyl and carboxyls. At moderate ageing states, a small band at 6.9 – 7.1 ppm indicates the formation of phenolic structures. The HNMR spectra of fresh naphthalene SOA show several singlets in the aromatic region, indicating a diversity of individual compounds occurring in relatively high concentrations,

while moderately aged SOA show mainly the two singlets of phthalic acid. All spectra contain signals at lower chemical shifts with respect to the aromatics, mainly between 3.5 and 6.0 ppm (with the interference of the partly suppressed peak of water in the middle): several functional groups can give rise to these bands, including alkoxy-, peroxy-, esters, hemiacetals and acetals, and vinyls.

The spectrum of the most aged sample (NAPTH 1), which is also the one with the lowest SOA concentration,
shows an interference from α-pinene SOA in the aliphatic region. This feature is likely due to experimental setting contamination from a previous α-pinene SOA experiment. Despite the low naphthalene SOA concentration, a broad aromatic band between 6.5 and 8.5 ppm and the same signals found between 3.5 and 6.0 ppm seen in the samples with a medium O/C ratio are still visible in this most aged naphthalene SOA spectrum. However, the band from oxygenated functional groups between 3 and 4.5 ppm becomes relatively
more intense with respect to aromatics compared to SOA samples of smaller ageing state. Compared to α-pinene SOA, the [1]H-NMR fingerprint of naphthalene SOA appears less sensitive to variations in the OH exposure between the low and the medium level of exposure. More substantial changes can be found for the most oxidized sample, which are only partly visible due the low signal-to-noise ratio of the spectrum.

**3.1.3 NMR fingerprints of non-IEPOX isoprene SOA**

The samples of isoprene SOA were obtained from the same OH exposure (corresponding approximately to a "medium" exposure in the α-pinene and naphthalene experiments) and differed only for collection time and sample quantity loaded on the filter. The isoprene SOA [1]H-NMR spectra profiles were all very similar (an
example is provided in Figure 3). The comparison with literature data (Budisulistiorini et al. 2015) led to the unambiguous identification of 2-methyltetrols, clearly responsible for the two singlets at 1.12 ppm (methylic H atoms of methylerythritol) and 1.13 ppm (methylic H atoms of methylthreitol) and for a series of multiplets between 3.4 and 3.9 ppm. Methylerythritol is more abundant (60% of the sum of the two, as an average between two samples extracted in water and three extracted in methanol) than methylthreitol. The spectra
show the occurrence of only two diastereomers among the possible four ones (González et al. 2011), indicating that the formation of methyl-tetrols is stereoselective, as already proposed by Cash et al. (2016) on the basis of a theoretical analysis of the IEPOX chemistry, and in contrast with the conclusions of González et al. (2011) claiming that methyltetrols are produced in laboratory conditions only in racemic mixtures. The two methyltetrols account for 65% of the total [1]H-NMR signal, the rest being characterized by broad
background signal with very few sharp resonances, indicating that the isoprene SOA samples are composed mainly of methyl-tetrols together with a significant amount of mass composed of a very complex mixture of products. The unresolved background resonances are located below the peaks of the methyltetrols, suggesting that the complex mixtures (which can include also oligomeric species) encompass molecular species (or monomers) similar to methyltetrols (at least in their C-H backbone). However, the range of chemical shifts
of the background bands characterize molecular species with more electronegative groups (leading to more de-shielded H atoms) than methyltetrols: the band of methylic protons extends to 1.7 ppm (respect to 1.12-1.13 of methyltetrols) and the band of alcoxyl groups (HC-O) extends to 4.3 ppm. The results of Liu et al. (2016), indicating that non-IEPOX isoprene SOA include peroxide-equivalents of methyltetrols, are in agreement with these findings. However, Riva et al. (2016) reported only peroxides for non-IEPOX SOA in
unseeded experiments and no methyl-tetrols. It is possible that peroxides decomposed to tetrols in our filter samples during collection downstream the PAM or afterwards during storing. The actual stability of isoprene hydroperoxides in the aerosol itself is largely uncertain, therefore the discrepancy between the findings presented in this study and the results of Riva et al. (2016) cannot be clarified at this stage. In addition, neither Liu et al. (2016) and Riva et al. (2016) reported the presence of carboxylic or keto groups, while our data
clearly indicate that these (and/or other acyl groups) are found in the unresolved mixtures of non-IEPOX isoprene SOA, and are responsible for the signal band between 2.0 and 2.6 ppm. Still, this band is much less intense than that of alcoxyls, which is opposite to what observed for α-pinene SOA where acyls are by far

the main oxygenated aliphatic functional group. Thus, $^1$H-NMR spectroscopy provides distinct fingerprint for isoprene and monoterpene SOA.


## 3.2 HPLC results

The HPLC analysis of fresh α-pinene SOA extracts shows the presence of compounds unretained by ion-exchange columns (neutral compounds) or weakly retained (mono- and di-acids) with a small contribution from compounds having a high retention factor (polyacids, PA, or HULIS), in agreement with previous
results obtained from α-pinene SOA samples generated in environmental chamber experiments (unpublished data). It should be noted, however, that the chromatographic analysis of SOA compounds in water extracts generally does not allow to recover high-molecular weight organic oligomers susceptible to hydrolysis reactions (e.g., polyacetals, Kroll and Seinfeld 2008). The HULIS determined by our method are essentially only the non-hydrolyzable ones, stable in aqueous solutions. The HULIS content increases only moderately
with ageing, while the yield/fraction of di-acids increases significantly with respect to mono-acids and neutral/basic compounds (Figure 4). With increasing photochemical age, the total organic carbon (TOC) mass fractions of mono-acids, di-acids and HULIS increase from 20% to 34% and 7% to 16%, respectively, whereas the mass fraction of neutral compounds decreases from 19% to 9% (Figure S3). These results are in qualitative agreement with the known chemistry of α-pinene SOA, in which mono- and dicarboxylic acids
are the most characteristic condensable 1$^{st}$ generation products (Jenkin et al., 2000; Jaoui and Kamens 2001) while tricarboxylic acids such as 3-methyl-1,2,3-butanetricarboxylic acid or pinyl-diaterpenyl ester (Szmigielski et al., 2007; Yasmeen et al., 2010) are present in lesser amounts and can contribute the observed concentrations of HULIS in this study.

The HPLC fractionation of naphthalene SOA (Figure 5) shows that fresh samples are characterized by a
mixture of neutral compounds and mono- and di-acids, completely consistent with the molecular compositions reported in the literature (Lee and Lane 2009). However, a net increase in acidic compounds with photochemical age can be clearly observed. However, the HULIS content, initially small, increases substantially and progressively with ageing. With increasing photochemical age, the TOC mass fraction of mono- and di-acids decreases from 33% to 18% and from 34% to 33% respectively, while the fraction of
PA/HULIS increases from 11% to 30% (Figure S4). This is the first evidence of HULIS formation (determined with the ion-exchange method) in laboratory-generated SOA. The formation of polyacidic molecules with three or more carboxylic groups implies the opening of the second aromatic ring in the naphthalene precursor backbone, and/or oligomerization reactions. In both cases, products of such oxidation reactions cannot be explained by current naphthalene SOA molecular speciation studies (e.g. Kautzman et
al., 2010).

Finally, the HPLC analysis of isoprene SOA shows that neutral compounds (NC) were dominant in all sample extracts (Figure 6). NC accounted for 59% of the TOC content of the sum of the HPLC fractions. The second most abundant fraction (34%) are the mono-acids, while diacids accounted for a much smaller fraction of TOC, and polyacids were almost absent (ca. 1% of the TOC). The dominance of NC is consistent with the
high yield methyltetrols and their analogues (see Section 3.1.3). Assuming that the distribution of NMR functional groups approximately reflects their carbon content, methyltetrols (accounting for 65% of the total NMR signal) can account for the whole of the HPLC neutral compounds, and, as a corollary, the complex mixtures of products detected by unresolved bands by NMR spectroscopy correspond to the mono- and di-acids in HPLC. As already noticed in the previous section, the NMR analysis shows indeed the occurrence
of acyl groups which indicate/support the presence of carboxylic acids. We cannot exclude, however, possible misclassification of some neutral compounds into the monoacid fraction, as already observed for some dicarbonyls (Decesari et al. 2005). It is worthwhile to clarify that the definition of chemical classes is based uniquely on the retention factor on the anion-exchange, and therefore sensitive to chromatographic secondary interactions and to chromatographic conditions. Such (unwanted) effects can explain the difference in
speciation between samples Pin#2 and Pin#3, both obtained at medium oxidation state (Figure S3). Pin#3

was injected at significantly higher concentrations than the other samples (about 3 times than Pin#2), which can have caused the elution of some acidic compounds along with the unretained NC fraction. The results of HPLC analysis of this particular sample are therefore excluded from the following discussion.

## 4 Discussion and conclusions

In this section, the NMR and HPLC results obtained for the isoprene, α-pinene and naphthalene SOA systems are compared with ambient OA samples. First, we investigated the similarity between the [1]H-NMR spectral profiles of SOA with those "typical" of ambient non-biomass-burning WSOC. For this purpose we used one sample of PM1 collected during the 2012 PEGASOS field campaign (Sandrini et al., 2016) in the rural Po Valley (Italy) which can be considered representative for a continental rural "near-city" site (according to the criteria of Putaud et al., 2010). A second PM1 sample was collected at a rural site in the State of Rondônia (Brazil) during the 2002 SMOCC field campaign, and, more precisely, during the early rainy season, when local biomass burning sources had largely ceased and the organic composition of submicron particles was dominated by biogenic emissions (Decesari et al. 2006; Tagliavini et al., 2006). The ambient WSOC and laboratory SOA spectra were binned to 400 points in order to remove the variability in chemical shifts due to, e.g., different pH conditions during the analyses of the samples. Figure 7 shows the correlation between the SOA spectra and the reference spectra of ambient WSOC: the α-pinene SOA and naphthalene SOA spectra were compared to the Po Valley WSOC sample, while the isoprene SOA spectra were compared to the Amazonian sample. There is good correlation ($0.62 < r < 0.92$) between the NMR spectra of α-pinene SOA, at all oxidation levels, with the spectrum of the Po Valley PM1 sample. This finding is in line with modelling results and previous experimental findings indicating that the organic composition in northern Italy in the summertime is dominated by biogenic SOA (Bessagnet et al., 2008; Gilardoni et al., 2011). A moderate positive correlation was also found between the spectra of isoprene SOA and of the PM1 sample from rural Brazil ($r = 0.52, 0.54$). It should be noted that the relative humidity at this pasture site is variable during the day and very high overnight during the rainy season, based on the meteorological data presented by Betts et al. (2009). Therefore, biogenic aerosols are expected to include also isoprene SOA forming through the IEPOX route (Hu et al., 2015), which is not accounted for by our laboratory experiments. Finally, the naphthalene SOA spectra exhibit zero or negative correlations with the Po Valley WSOC spectrum ($-0.15 < r < -0.02$). This result is somewhat expected if we consider that ambient water-soluble aerosols are characterized by acyl functional groups (HC-C=O) in higher concentrations with respect to alcoxyl groups (HC-O) and by a smaller aromatic content (Decesari et al., 2007), with a functional group pattern that is well reproduced by α-pinene SOA and not by naphthalene SOA. Clearly, naphthalene SOA *alone*, with hydrogen-to-carbon (H/C) ratio less than 0.9 due to relatively high aromatic content (Lambe et al. 2011, 2015), does not mimic ambient OA *bulk* composition. It should be noted, however, that naphthalene and other polyaromatic hydrocarbons are co-emitted with many other anthropogenic IVOCs and VOCs including aliphatic compounds in the real atmosphere; therefore the contribution of naphthalene SOA could be simply masked by the contribution of aliphatic IVOC SOA in the [1]H-NMR spectra of ambient WSOC. When limiting the correlation analysis of Figure 7 to the aromatic and vinyl region of the spectra (> 6 ppm), Pearson $r$ coefficients of 0.49 to 0.58 are found between the naphthalene SOA and the ambient WSOC spectra, while small values (between -0.2 and 0.4) are found for the α-pinene and isoprene SOA. This result suggests that SOA produced from naphthalene or similar precursors, including many other ring-retaining oxidation products (Section 3.1.2) can explain the presence of aromatic moieties in ambient water-soluble aerosols in areas not affected by biomass burning emissions.

When considering the full spectral range, the H-NMR spectra of α-pinene SOA most closely mimic the functional group distributions of the ambient WSOC sample obtained in PEGASOS (Fig. S1). Interestingly, similarity between H-NMR spectra of α-pinene SOA and Po Valley WSOC increases with increasing photochemical age. For the most oxidized α-pinene SOA samples, the functional group composition, characterized by polysubstituted aliphatic compounds rich of acyls (carboxylic or keto groups), overlaps well with that of ambient WSOC. A good fit between α-pinene SOA and ambient WSOC mass spectral features

is already achieved at medium oxidation conditions, in agreement with the results of Lambe et al. (2011) showing that the correlation between the AMS spectra of PAM-generated α-pinene SOA with the spectra of ambient OOA increases up to an exposure of 1 x $10^{12}$ OH molec $cm^{-3}$*s and remains rather stable afterwards. These results suggest that NMR-traced ageing processes reflect the same chemical mechanisms already studied using high-resolution AMS techniques. However, the correlation coefficients shown in Figure 7 for the NMR spectra of α-pinene vs. ambient WSOC ($r^2$ = 0.39 to 0.84) are smaller than those between the HR-ToF-AMS spectra of PAM-generated SOA vs. ambient OOA ($r^2$ = 0.7 to 0.9) (Fig. 9 in Lambe et al., 2011). Apparently, the AMS features of ambient OA are more easily reproduced by PAM experiments than the NMR composition, or, in other words, NMR spectroscopy exhibits a greater selectivity for the OA components than AMS. Specifically, $^1$H-NMR spectroscopy was able to resolve significant changes in composition of α-pinene SOA with photochemical ageing in great detail, especially at an OH exposure of ~1.1 * $10^{12}$ molec OH /$cm^3$ s equivalent to multiple days of atmospheric ageing. It should be noted, finally, that a comparison of the AMS and NMR techniques with respect to their ability to trace chemical ageing in laboratory SOA and ambient oxidized aerosols is challenged by the incomplete overlap between the classes of organic compounds contributing to OOA and to WSOC (Xu et al., 2017).

A comparison of fresh and aged SOA with ambient WSOC samples with respect to the HPLC fraction distributions is reported in Figure 8. The distribution of neutral vs. acidic classes of compounds in ambient WSOC refers to the average of the samples collected at the rural background station of Cabauw in the Netherlands (Paglione et al. 2014b). The station is located downwind from anthropogenic sources and biogenic emissions (terpenes from deciduous forests) over a large sector of north-west Europe (Henne et al., 2010). The HULIS contribution in these samples varied between 15 and 20%, in line with previous results obtained in the Po Valley (Mancinelli et al., 2007), but lower than in biomass burning aerosol samples (Decesari et al., 2006). The α-pinene SOA generated in the PAM reactor at high photochemical age and the fresh naphthalene SOA are characterized by a HULIS amount similar to that of Cabauw samples, while the polyacidic content of aged naphthalene SOA is higher than in the ambient samples. In the real atmosphere, naphthalene is co-emitted with many other reactive VOC and IVOC with potentially very diverse HULIS formation yields, therefore the results presented in Figure 8 do not necessarily mean that the chemical composition of ambient OA in Cabauw is better described by the monoterpene chemistry rather than by anthropogenic IVOC oxidation. On the other hand, these results demonstrate that laboratory experiments of SOA formation can generate complex mixtures of products with the same chromatographic properties of HULIS provided a sufficient extent of photochemical aging using the PAM reactor or related techniques. The HULIS fraction of WSOC is in fact proportional to the AMS f44 for SOA (integrated over the filter sampling times) (Figure 9) irrespectively of precursor type. Therefore, the formation of polycarboxylic acids determined by the HPLC technique follows the same positive trend in concentrations of the AMS proxy for C(O)OH groups with increasing OH exposure. This is in contrast with the numerous observations of rapid formation of SOA oligomers during reaction chamber experiments (Kalberer et al., 2006; Reemtsma et al., 2006), indicating that oligomers do not account for chromatographically-defined HULIS. A survey of the laboratory studies on the formation of humic material in secondary aerosol shows that evidence for the formation of polycarboxylic acids comes from the reaction of phenolic compounds in the presence of particulate water (Hoffer et al., 2004), while little is known for unseeded, dry gas-to-particle formation experiments. With the exception of the very preliminary data reported by Baltensperger et al. (2008), our results – to our best knowledge – are the first showing the occurrence of HULIS *sensu strictu* in monoterpene and aromatic hydrocarbon SOA, and these HULIS are clearly shown to be a product of photochemical ageing.

In conclusion, we observed that OA ageing reactions in the PAM reactor produces water-soluble compounds of high complexity but with spectroscopic and chromatographic properties that converge towards those characteristic of ambient OA. Specifically, a good correlation between ambient HPLC/HNMR samples and aged α-pinene SOA was observed in respect to HULIS content and NMR functional group distribution, while aged aromatic IVOC SOA show clear potential for HULIS formation. The isoprene SOA samples do not show compositional features with a clear overlap with those of ambient WSOC obtained in the Cabauw and

Po Valley samples that are representative of continental polluted atmospheres, but should serve as useful
reference spectra for future studies/environments impacted by non-IEPOX isoprene SOA.

**Acknowledgements**

A.T. Lambe and P. Massoli acknowledge support by the Atmospheric Chemistry Program of the U.S. National Science Foundation under grants AGS-1536939, AGS-1537446 and by the U.S. Office of Science
(BER), Department of Energy (Atmospheric Systems Research) under grants DE-SC0006980 and DE-SC0011935. We thank Manjula Canagaratna (ARI), Douglas Worsnop (ARI), Timothy Onasch (BC/ARI) and Paul Davidovits (BC) for helpful discussions. S. Decesari, S. Gilardoni, M. Paglione and N. Zanca acknowledge funding from the European FP7 project BACCHUS (grant agreement No. 49 990 603445). Dr. Fabio Moretti, formerly at the Department of Chemistry of the University of Bologna, Prof. Andrea Mazzanti
and Dr. Alessandra Petroli of the Department of Industrial Chemistry of the University of Bologna are also greatly acknowledged for support with the NMR analyses.

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

# Tables

| Sample | Oxidation level | PAM Lamp voltage (V) | OH Exposure (molec/cm$^3$*s) | Collection Time (h) | average AMS OM concentration (RIE*CE=1.4) (μg/m$^3$) | OM on filter based on AMS (RIE*CE=1.4) (μg) |
|---|---|---|---|---|---|---|
| *α-pinene* | | | | | | |
| Pin#1 | High (f44 = 0.24) | 110 | 2.10x10$^{12}$ | 18.5 | 46.1 | 384 |
| Pin#2 | Med. (f44 = 0.11) | 75 | 1.10x10$^{12}$ | 3.2 | 151.6 | 221 |
| Pin#3 | Med. (f44 = 0.11) | 75 | 1.10x10$^{12}$ | 20.2 | 14.1 | 129 |
| Pin#5 | Low (f44 = 0.05) | 30 | 2.00x10$^{11}$ | 20.5 | 7.5 | 69 |
| Pin#6 | Low (f44 = 0.05) | 30 | 2.00x10$^{11}$ | 7.1 | 50.3 | 161 |
| *Naphthalene* | | | | | | |
| Nap#1 | High f44 = 0.30) | 110 | 2.10x10$^{12}$ | 19.7 | 31.3 | 277 |
| Nap#2 | Med. (f44 = 0.19) | 75 | 1.10x10$^{12}$ | 7 | 55.3 | 174 |
| Nap#3 | Low (f44=0.084) | 30 | 2.00x10$^{11}$ | 6.6 | 16.9 | 50 |
| Nap#4 | Med (f44 = 0.20) | 75 | 1.10x10$^{12}$ | 6.6 | 42.9 | 127 |
| Nap#5 | Low (f44 = 0.074) | 30 | 2.00x10$^{11}$ | 15.8 | 34.7 | 246 |
| *blanks* | | | | | | |
| Blk#1 | | 110 | 2.10x10$^{12}$ | 6.1 | 0 | 0 |
| Blk#2 | | 110 | 2.10x10$^{12}$ | 23.2 | 0 | 0 |
| Blk#3 | | 110 | 2.10x10$^{12}$ | 6.1 | 0 | 0 |


**Table 1. PAM experimental conditions for naphthalene and α-pinene SOA ageing studies.**


| Sample | Oxidation level | PAM Lamp voltage (V) | OH Exposure (molec/cm$^3$*s) | Collection Time (h) | average AMS OM concentration (RIE*CE=1.4) (µg/m$^3$) | OM on filter based on AMS (RIE*CE=1.4) (µg) |
|---|---|---|---|---|---|---|
| *Isoprene* | | | | | | |
| Iso#1 | Med (f44 = 0.046) | 60 | 7.8*10$^{11}$ | 3.7 | 409 | 651 |
| Iso#2 | Med (f44 = N.A.) | 60 | 7.8*10$^{11}$ | 2.8 | 575 | 700 |
| Iso#3 | Med (f44 = 0.039) | 60 | 7.8*10$^{11}$ | 2.2 | 678 | 656 |
| Iso#4 | Med (f44 = 0.037) | 60 | 7.8*10$^{11}$ | 2.9 | 551 | 684 |
| Iso#5 | Med (f44 = 0.040) | 60 | 7.8*10$^{11}$ | 16.1 | 685 | 4959 |
| Iso#6 | Med (f44 = 0.066) | 60 | 7.8*10$^{11}$ | 3.9 | 767 | 1280 |
| Iso#7 | Med (f44 = 0.059) | 60 | 7.8*10$^{11}$ | 3.8 | 593 | 986 |
| *Blanks* | | | | | | |
| Blk#1 | | 60 | 7.8*10$^{11}$ | 15.9 | 0 | 0 |
| Blk#2 | | 60 | 7.8*10$^{11}$ | 3.1 | 0 | 0 |
| Blk#3 | | 60 | 7.8*10$^{11}$ | 16.3 | 0 | 0 |

**Table 2 PAM experimental conditions for isoprene SOA ageing studies.**





# Figures

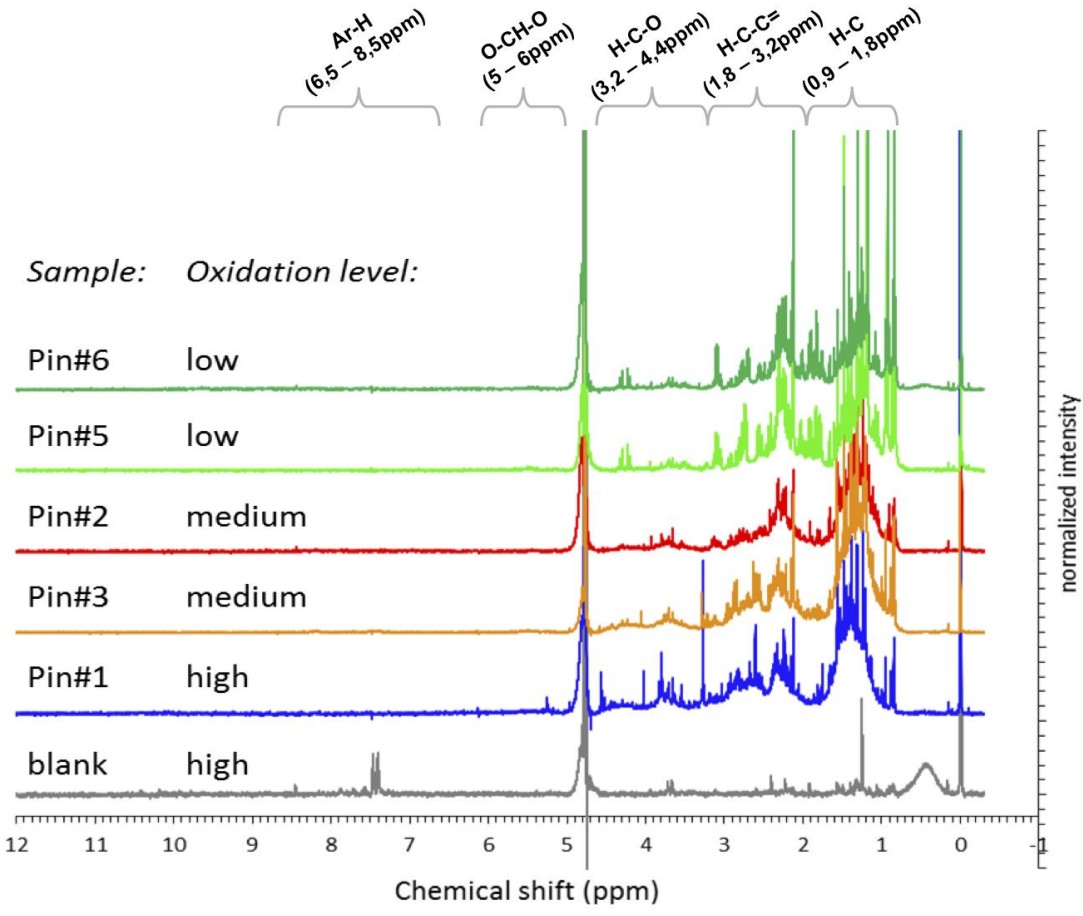

**Figure 1. ¹H-NMR spectra of α-pinene SOA as a function of increasing photochemical age in the Potential Aerosol Mass (PAM) oxidation flow reactor. The sharp singlet at zero ppm represents the internal standard (Tsp-d4), while the broad peak at 4.8 ppm is the – partly instrumentally suppressed – HDO peak. The sharp singlets between 0.9 and 2.2 ppm in the fresh SOA samples (Pin#5 and Pin#6) are genuine bands of the samples and were identified as methyl groups of pinic and pinonic acid.**


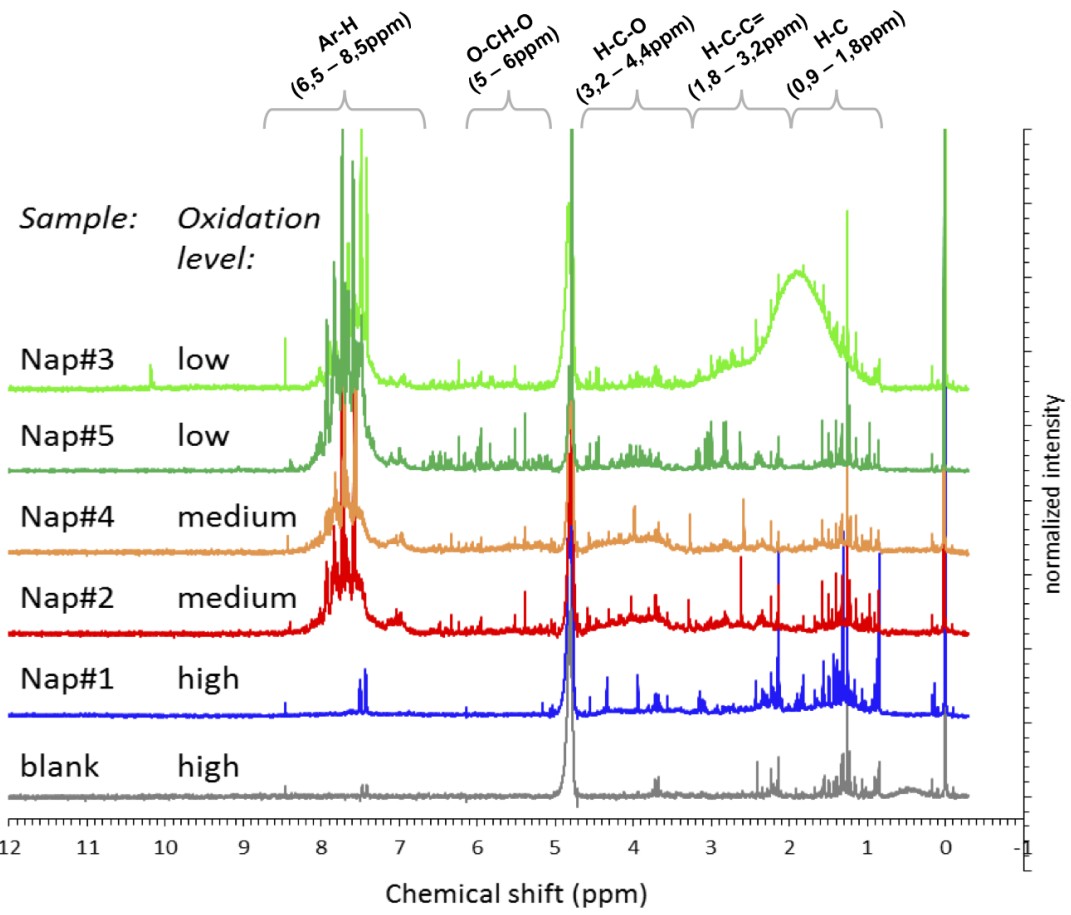

**Figure 2. ¹H-NMR spectra of naphthalene SOA as a function of increasing photochemical age in the PAM reactor. The sharp singlet at zero ppm represents the internal standard (Tsp-d4), while the broad peak at 4.8 ppm is the – partly instrumentally suppressed – HDO peak.**


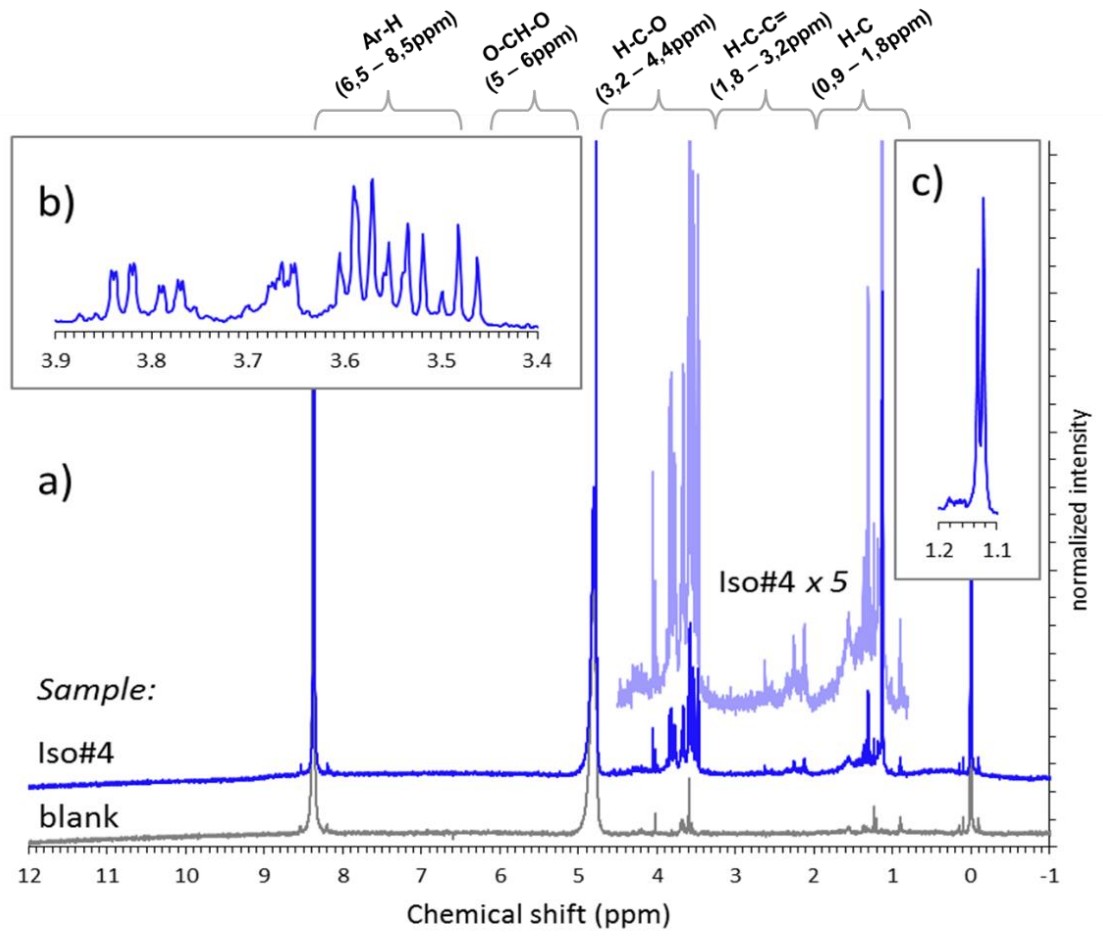

**Figure 3. Panel a: $^1$H-NMR spectrum of isoprene SOA generated in the PAM reactor at an OH exposure of 8*10$^{11}$ molec cm$^{-3}$ sec. The bottom trace shows the same spectrum with enlarged the broad background bands. Panel b and c show the methyltetrols resonances between 3.4 and 3.9 ppm and 1.12-1.13 ppm, respectively**


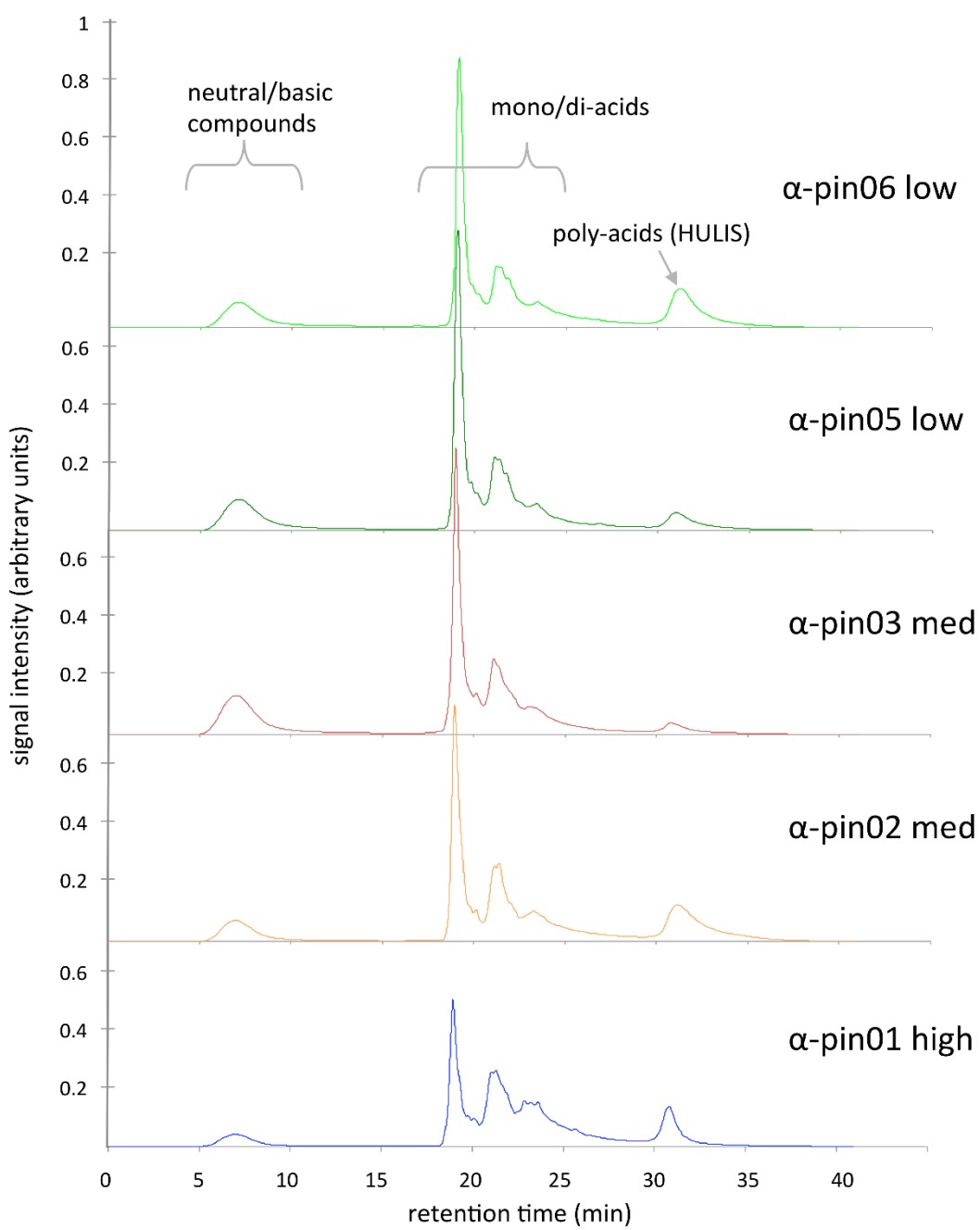

**Figure 4.** HPLC chromatograms of α-pinene SOA water extracts. Chromatographic features are grouped into neutral, mono-/di-carboxylic acid, and polycarboxylic acid classes based on their affinity for the column phase. Sample identifications are provided in Table 1.


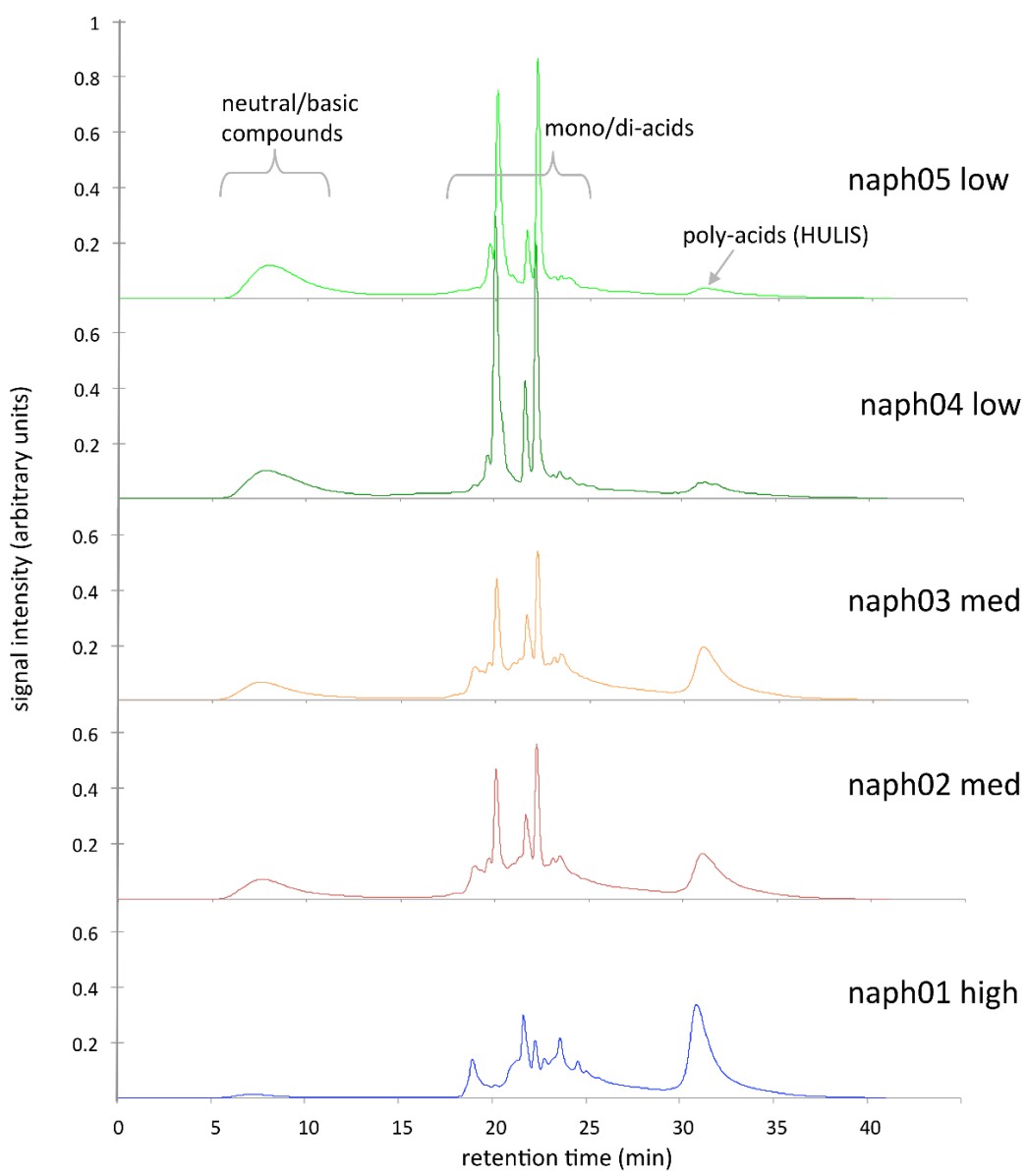

**Figure 5. HPLC chromatograms of naphthalene SOA water extracts. Chromatographic features are grouped into neutral, mono-/di-carboxylic acid, and polycarboxylic acid classes based on their affinity for the column phase. Sample identifications are provided in Table 1.**


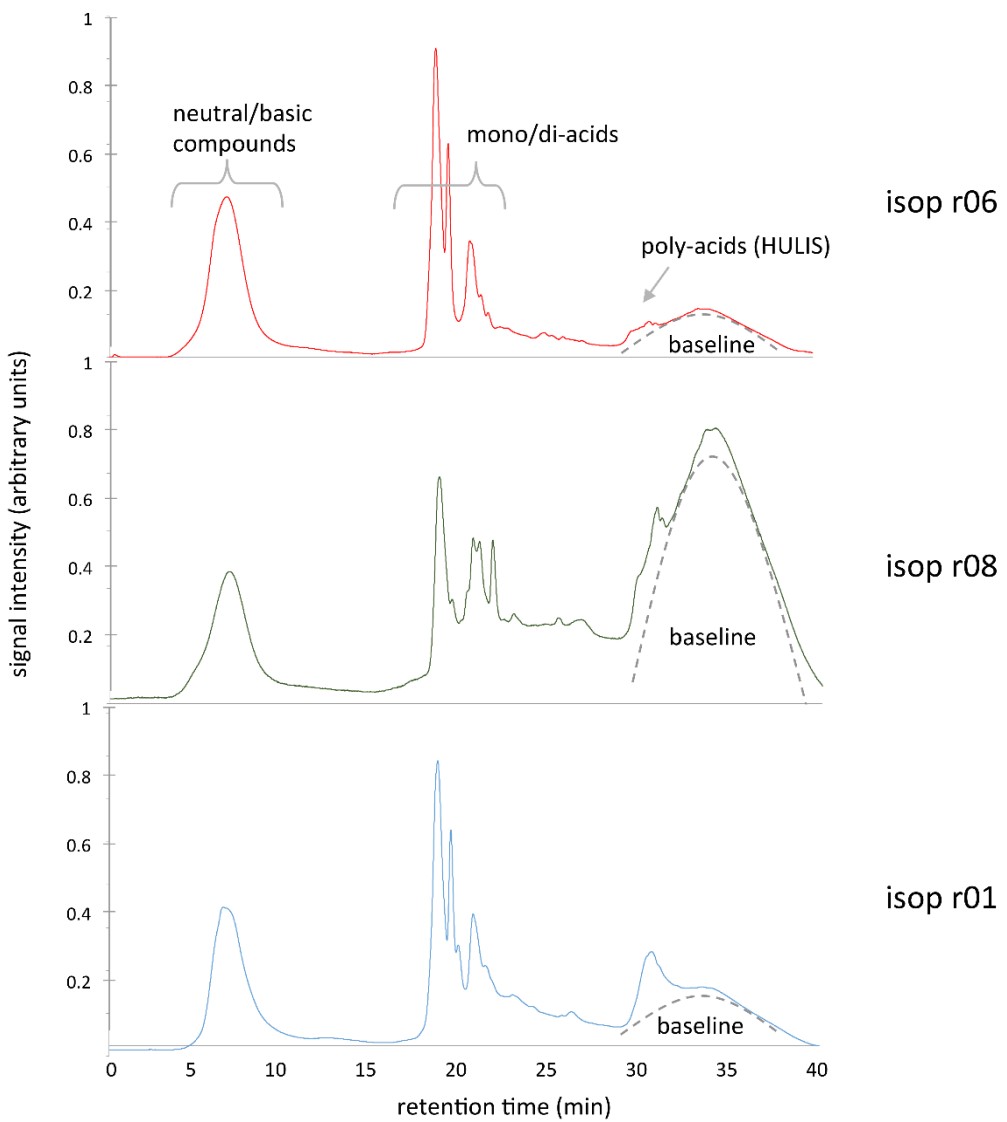

Figure 6. HPLC chromatograms of isoprene SOA water extracts. Chromatographic features are grouped into neutral, mono-/di-carboxylic acid, and polycarboxylic acid classes based on their affinity for the column phase. Sample identifications are provided in Table 2.

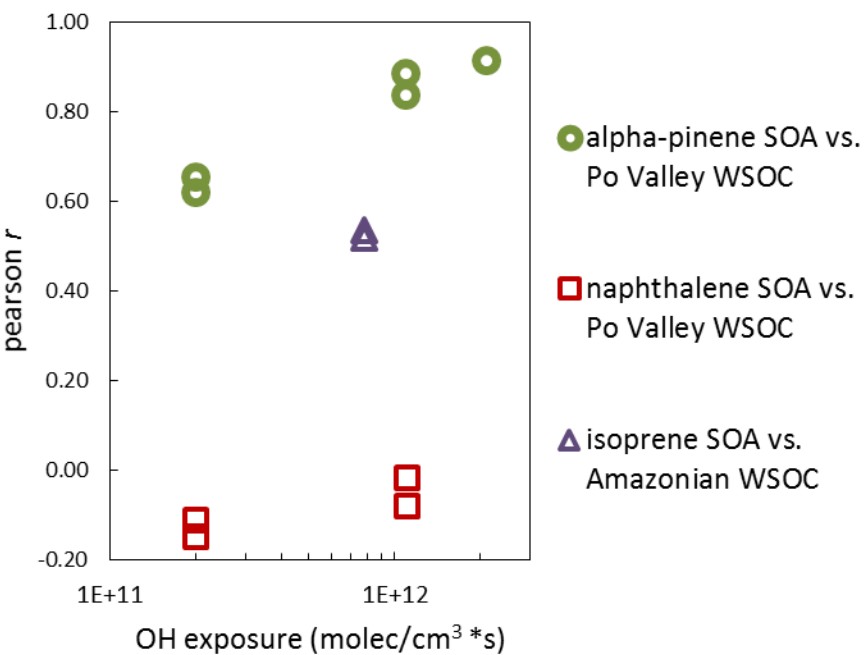

**Figure 7. Pearson correlation coefficient between H-NMR spectra of PAM-generated SOA and ambient PEGASOS WSOC.**

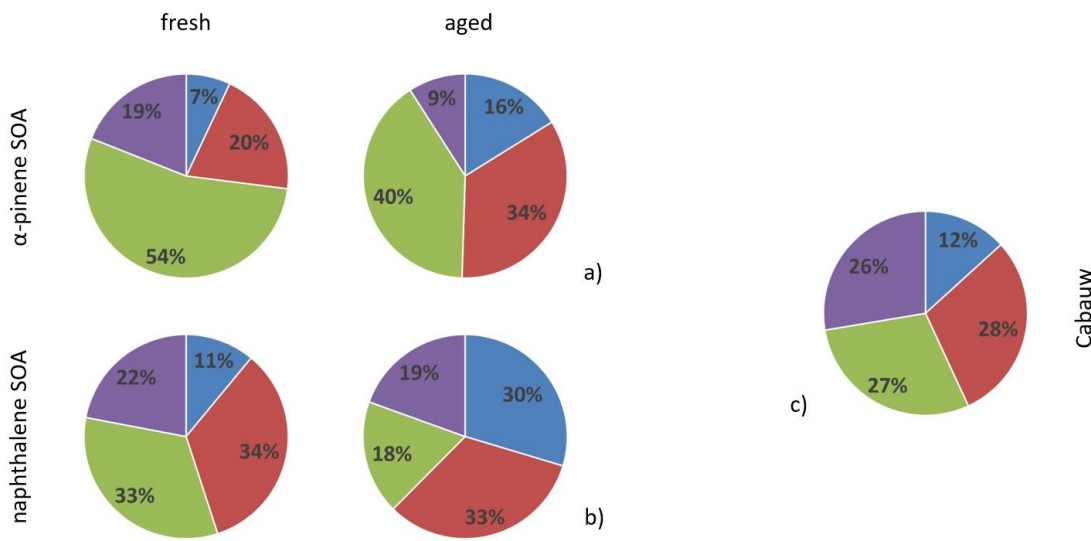

**Figure 8. Distribution of HPLC fractions (total recovered TOC content = 100%) for α-pinene SOA (panel a), naphthalene (panel b), and for ambient OA sampled in Cabauw, Netherlands (panel c).**


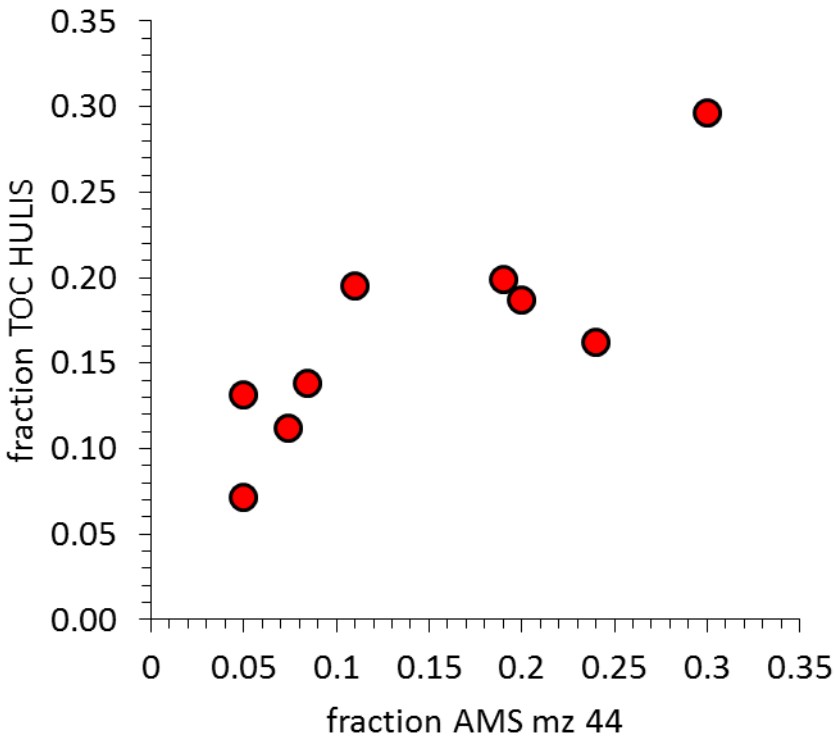

**Figure 9. Correlation plot between the AMS f44 of SOA and the HULIS fraction of HPLC-eluted WSOC.**