# Peer review of "Characterizing source fingerprints and ageing processes in laboratory-generated secondary organic aerosols using proton-nuclear magnetic resonance (1H-NMR) analysis and HPLC HULIS determination."

_Atmospheric Chemistry and Physics, 2017_

## Referee Comment (RC1) · Anonymous Referee #1 · 14 Mar 2017

1. General comments: This article reports a very interesting study on the compositional features of secondary organic aerosol (SOA) samples generated using a Potential Aerosol Mass (PAM) oxidation flow reactor using organic precursors that are well-established source fingerprints: alpha-pinene (most studied biogenic VOC), naphthalene (proxy for anthropogenic aromatics), and isoprene (most abundant biogenic VOC). The generated SOA samples were characterized/analyzed by means of proton nuclear magnetic resonance (1H NMR) spectroscopy and ion-exchange chromatogra-

phy coupled to a UV detector (on-line) and a TOC analyzer (off-line). SOA is one of the least understood constituents of fine aerosol particles; current widely-used models cannot predict its atmospheric loadings, oxidation state, or even the nature of the atmospheric ageing processes. Understanding, characterize, and (semi-)quantify the effects of SOA formation and ageing is challenging because it requires a different framework to capture and describe the continuous evolution of the structural features of organic compounds. Nonetheless, if those aims are attained, the outcomes will be very important for the atmospheric chemistry modelling community. In this Reviewer opinion, the topic of this paper is relevant to the journal's interests and will be of interest to readers. Studies such as this one are needed to advance our understanding on SOA oxidative aging mechanisms at ambient conditions. All in all, the quality of the measurements is excellent and the presentation and discussion of data is good. Nevertheless, this Reviewer has identified some issues requiring further clarification from the Authors. I recommend publication of this study after the Authors consider the specific comments (below).

2. Specific comments: - Section 3.1.1, lines 198-200: Although there is a change in the intensity of NMR peaks between 1 and 3 ppm of alpha-pinene SOA with photochemical age, the 1H NMR spectra of alpha-pinene SOA at medium ageing still exhibits resonance at the same chemical shift regions of alpha-pinene SOA with low oxidation level. Therefore, I do not think that there is a sharp change of NMR fingerprinting of alpha-pinene SOA already at medium ageing.

- Section 3.1.2, line 218: The Authors state that "moderately aged SOA show mainly the two singlets of phthalic acid" in the aromatic region. The singlets of phthalic acid should appear/resonate at approximately 8.1 ppm. This assignment is unclear in the spectra of naphthalene SOA (Figure 2) due to the presence of a broad resonance at approximately 7.2-8.3 ppm.

- Section 3.1.2, lines 218-221: Besides exhibiting NMR peaks between 3.5 and 6.0 ppm, all spectra also exhibit noticeable NMR peaks between 1 and 2 ppm. Could

these NMR resonances be still a consequence of the presence of colloidal hydrophobic material in solution, or they could be attributed to aliphatic structures derived from the ageing process?

- Section 3.1.2, lines 224-229: In my opinion, the broad aromatic band between 6.5 and 8.5 ppm is totally indiscernible in the spectrum of the most aged naphthalene SOA sample. I am also compelled to disagree when the Authors conclude that "there is no clear trend in the formation/disappearance of aromatic and aliphatic bands with ageing". In my opinion, the 1H NMR spectrum of the most aged sample clearly indicate the disappearance of resonance in the aromatic region, whereas a few NMR peaks resonate in the aliphatic region (chemical shifts = 3.5 and 5.0 ppm). The Authors should include additional explanations to support their statement/conclusion.

- Section 3.2, lines 275-276: The chromatograms in Figure 5 suggest that the signal intensity of the chromatographic peaks corresponding to mono- and di-acids increases with increasing photochemical age, which seems to contradict the statement "that the TOC mass fraction of mono- and di-acids decreases from 33% to 18%" with increasing photochemical age. Besides, in lines 273-374, the Authors conclude that "a net increase in acidic compounds with photochemical age can be clearly observed". Additional explanations should be provided to clarify these apparent conflicting conclusions.

- Section 3.2, line 282: the abbreviature "NC" for "neutral compounds" should have been previously defined in line 259.

- Section 4, lines 303-305: In Figure 7, it is unclear which marker correspond to ambient PEGASOS WSOC for the different oxidation levels.

- Section 4, lines 321-322: In the sentence "the H-NMR spectra of alpha-pinene SOA most closely mimic the functional group distributions of the ambient WSOC sample obtained in PEGASOS", are the Authors referring to the spectrum in Figure S1? If so, please redirect the reader to Figure S1.

- Section 4, lines 331-335: The Authors conclude that the correlation coefficients shown in Figure 7 for the NMR spectra of alpha-pinene vs. ambient WSOC are smaller than those between the HR-ToF-AMS spectra of PAM-generated SOA vs. ambient OOA reported in reference Lambe et al. (2011). Could this difference be explained by the fact that Lambe et al. (2011) is focusing on ambient OOA, which is a fraction of total organic aerosols (OA) and a proxy for SOA, whereas in this study the Authors are comparing the NMR spectra of alpha-pinene with those from the whole WSOC fraction which probably also includes a small fraction of primary OA? Secondly, I did not find Figure 9 in reference Lambe et al. (2011).

3. Technical corrections: - Section 2.1, Tables 1 and 2: There is some information missing from the "Oxidation Level" column for Pin#3 (Med.) and Pin#5 (Low) in Table 1, and for Iso#2 (med) in Table 2. Could you please also clarify the meaning of "f44" in these Tables? Is this corresponding to the m/z marker of COOH formation withdrawn from the AMS data? If so, this should be clarified in the manuscript.

- Section 2.2, line 144: Figure S1 shows the 1H NMR spectrum of an ambient WSOC sample collected in San Pietro Capofiume and not the 1H NMR spectra of methanol extracts from isoprene SOA samples. Instead, it could be Figure S2?

- Section 2.4, line 167: I did not find the full HPLC-UV-TOC analytical protocol in the Supplementary material.

4. References: Lambe A. T., Ahern A. T., Williams L. R., Slowik J. G., Wong J. P. S., Abbatt J. P. D., Brune W. H., Ng N. L., Wright J. P., Croasdale D. R., Worsnop D. R., Davidovits P., and Onasch T. B.: Characterization of aerosol photooxidation flow reactors: heterogeneous oxidation, secondary organic aerosol formation and cloud condensation nuclei activity measurements, Atmos. Meas. Tech., 4, 445–461, 2011.

---

## Referee Comment (RC2) · Anonymous Referee #2 · 24 Mar 2017

Review of "Characterizing source fingerprints and ageing processes in laboratory-generated secondary organic aerosols using proton-nuclear magnetic resonance (1H-NMR) analysis and HPLC-HULIS determination."

**General Comments:**

This manuscript describes the chemical analysis of laboratory-generated secondary organic aerosol (SOA) from isoprene, alpha-pinene, and naphthalene precursors. The authors generate SOA within a Potential Aerosol Mass (PAM) oxidation flow reactor, varying actinic flux to achieve various overall OH exposures, collect SOA on quartz fiber-filters, and then extract water soluble organics for analysis using 1H-NMR and HPLC. The authors find that with increased OH exposure 1H-NMR analysis shows further oxidized organic material being generated, consistent with spectroscopic features of known oxidation products from the studied systems. In comparison to 1H-NMR and HPLC analysis of ambient samples taken from the Po Valley, Italy, and Cabauw, Netherlands, the authors find that highly-oxidized alpha-pinene SOA 1H-NMR spectroscopic features and HULIS chromatographic features from similarly aged alpha-pinene and naphthalene most resemble that of the ambient samples.

Overall the manuscript is clearly written and the experimental setup and analysis straight forward. The results have potential to be of interest to ACP readers, though I think improvements are needed in discussing and considering the true atmospheric relevance of the current work. In particular, the inclusion of the isoprene results seem superfluous and of confusing atmospheric relevance because of potentially contradictory experimental conditions used and results obtained, and also because the selected ambient samples were not taken from areas notable for isoprene-dominated chemistry. Additional experimental methodology should be clarified as well. After specific and technical comments are addressed below, this manuscript may be suitable for publication in ACP.

**Specific comments:**

1. Lines 31-32: Please clarify the definition of the maximum yield %'s listed. Are these yields mass of HULIS generated per mass of precursor reacted away, similar to an SOA yield? If so, please provide quantification/calibration information. Rather, are they fraction of WSOC that is designated as HULIS (as presented in SI Figs. S3, S4)?
2. Lines 50-79: In light of the discussion here, the authors should provide additional context about the major SOA formation processes governing their particular experiment. Since the experiments are unseeded, what is the relative importance and timescale of nucleation, heterogeneous oxidation, and gas-particle partitioning in generating the characterized SOA? And how do these processes and timescales differ with ambient timescales even if similar total OH exposure is achieved?
3. Line 73: Consider including these additional references in discussing vapor-phase wall losses: (Matsunaga and Ziemann, 2010; Ye et al., 2016)
4. Lines 87-89: This statement does not seem fair or accurately written considering the authors cite work of Cavalli et al., 2006 later for comparing alpha-pinene SOA and plenty of 1H-NMR spectroscopy has been performed on SOA samples from urban areas e.g. (Cleveland et al., 2012). Do the authors want to reword the sentence to specify something specific about novelty of applying the technique to *laboratory-generated* SOA from known biogenic and anthropogenic SOA precursors?

5. Lines 151-153: Can the authors clarify if this choice in using the buffer has to do with the specific products expected from non-IEPOX isoprene chemistry that may be subject to chemical shift of the hydrogen atoms?

6. Interpretation of isoprene results in Section 3.1.3 NMR fingerprints of non-IEPOX isoprene SOA: In light of the experimental conditions being dry (according to lines 117) and unseeded, why do the authors attribute the spectroscopic features to 2-methyl tetrols as products in their experiments? These are typically observed as particle-phase products resulting from IEPOX route of SOA formation from isoprene, but the authors claim their conditions may resemble more of non-IEPOX route of SOA formation according to Krechmer et al., 2015 in lines 107-112 and Liu et al., 2016 in lines 249-250. Do the authors suggest that peroxide-equivalents of 2-methyl tetrols (formed via nucleation under their experimental conditions?) would have the exact same NMR fingerprint as 2-methyl tetrols? If so, this really weakens the use of NMR fingerprints with true molecular-level mechanistic understanding. The authors should reconcile their measurements and the mechanism more clearly.

7. Lines 279-281, Lines 355-361: The authors should clarify if water extraction methods of the organic material from filters can lead to aqueous processing and artificial formation of HULIS.

8. Line 298: The experimental setup is dry. How can the authors be certain that they are generating the same compounds in SOA formation/mimicking atmospherically relevant mechanisms from their chosen precursors and then comparing to WSOC from atmospheric samples?

9. Lines 346-349: While this may be true, Figure 8 seems to suggest that even just fresh naphthalene SOA has the closest WSOC breakdown of HPLC fractions with the ambient OA sample. This suggests that the experimental results are getting the right HPLC fractions, but not necessarily for the right composition/same mechanisms and for that matter the same oxidative aging. Considering the ambient sample is from a rural site, it also seems odd that the naphthalene WSOC breakdown is so close to that observed.

10. Please provide additional context for the ambient samples from Cabauw and Po Valley. Are the campaigns noted to be regions of high monoterpene emissions and anthropogenic (aromatic) emissions? They are both described as be rural sites, yet see comment 9 above.

11. Figures 1-3: Similar to the regions pointed out in HPLC results for neutral, mono/di-acids, HULIS, it would be helpful if there were similar labels pointing out characteristic bond features associated with the chemical shifts in the 1H-NMR spectra.

12. Figures 1-6: As plotted, it is sometimes difficult to really compare the intensities along an arbitrary y-axis with no units to compare peak heights and especially with peaks that extend into another sample's y-axis. Consider individual y-axes for each sample with relevant ticks or overlaying some samples' spectra?

13. Figure 9: Seems like the range of f44 is in quite good agreement with HULIS as shown in triangle plot of (Ng et al., 2011). What about f43? This would strengthen argument of associated HULIS with photochemical age and atmospheric relevance.

**Technical comments:**

1. Lines 39-40: Sentence is awkwardly worded.  Please revise, e.g. "In the mid 2000's the discovery that oxidized organic compounds dominate in concentration compared to that of primary organic compounds outside urban areas…"
2. Line 186: Spelling on "twards"
3. Line 296: Missing alpha symbol for alpha-pinene
4. Line 329: Insert "to" after "up".

**References:**

Cleveland, M. J., Ziemba, L. D., Griffin, R. J., Dibb, J. E., Anderson, C. H., Lefer, B. and Rappenglück, B.: Characterization of urban aerosol using aerosol mass spectrometry and proton nuclear magnetic resonance spectroscopy, Atmos. Environ., 54, 511–518, doi:10.1016/j.atmosenv.2012.02.074, 2012.

Matsunaga, A. and Ziemann, P. J.: Gas-Wall Partitioning of Organic Compounds in a Teflon Film Chamber and Potential Effects on Reaction Product and Aerosol Yield Measurements, Aerosol Sci. Technol., 44(10), 881–892, doi:10.1080/02786826.2010.501044, 2010.

Ng, N. L., Canagaratna, M. R., Jimenez, J. L., Chhabra, P. S., Seinfeld, J. H. and Worsnop, D. R.: Changes in organic aerosol composition with aging inferred from aerosol mass spectra, Atmos. Chem. Phys., 11(13), 6465–6474, doi:10.5194/acp-11-6465-2011, 2011.

Ye, P., Ding, X., Hakala, J., Hofbauer, V., Robinson, E. S. and Donahue, N. M.: Vapor wall loss of semi-volatile organic compounds in a Teflon chamber, Aerosol Sci. Technol., doi:10.1080/02786826.2016.1195905, 2016.

---

## Author Comment (AC1) · 4 Jul 2017

**Replies to Referee 1's comments.**

We thank the Referee for the through review. Below we copy his/her comments (in italics) and provide point-by-point replies.

*1. General comments: This article reports a very interesting study on the compositional features of secondary organic aerosol (SOA) samples generated using a Potential Aerosol Mass (PAM) oxidation flow reactor using organic precursors that are well-established source fingerprints: alpha-pinene (most studied biogenic VOC), naphthalene (proxy for anthropogenic aromatics), and isoprene (most abundant biogenic VOC). The generated SOA samples were characterized/analyzed by means of proton nuclear magnetic resonance (1H NMR) spectroscopy and ion-exchange chromatography coupled to a UV detector (on-line) and a TOC analyzer (off-line). SOA is one of the least understood constituents of fine aerosol particles; current widely-used models cannot predict its atmospheric loadings, oxidation state, or even the nature of the atmospheric ageing processes. Understanding, characterize, and (semi-)quantify the effects of SOA formation and ageing is challenging because it requires a different framework to capture and describe the continuous evolution of the structural features of organic compounds. Nonetheless, if those aims are attained, the outcomes will be very important for the atmospheric chemistry modelling community. In this Reviewer opinion, the topic of this paper is relevant to the journal's interests and will be of interest to readers. Studies such as this one are needed to advance our understanding on SOA oxidative aging mechanisms at ambient conditions. All in all, the quality of the measurements is excellent and the presentation and discussion of data is good. Nevertheless, this Reviewer has identified some issues requiring further clarification from the Authors. I recommend publication of this study after the Authors consider the specific comments (below).*

*2. Specific comments:*

*- Section 3.1.1, lines 198-200: Although there is a change in the intensity of NMR peaks between 1 and 3 ppm of alpha-pinene SOA with photochemical age, the 1H NMR spectra of alpha-pinene SOA at medium ageing still exhibits resonance at the same chemical shift regions of alpha-pinene SOA with low oxidation level. Therefore, I do not think that there is a sharp change of NMR fingerprinting of alpha-pinene SOA already at medium ageing.*

**REPLY:** There are indeed several specific resonances, some of which are known tracers of alpha-pinene oxidation, found in all alpha-pinene SOA samples, but their intensity decreases with ageing, especially between the fresh to the medium oxidation state samples. This shows up more clearly when using an y scale for Figure 1 illustrating the highest peaks (from pinic and pinonic acids in Pin#5 and Pin#6 samples) (see figure below). The integral of the resonance at 0.83 ppm arising from one of the two gem-methyls of pinonic acid changes from 2% and 3% of the total integral of the spectrum for samples Pin#5 and Pin#6, respectively, to 0.3% for all the other three samples. Therefore, the contribution of first generation oxidation products of alpha-pinene is one order of magnitude greater in fresh SOA than in the medium and high ageing state samples. This confirms that there is in fact a marked change in NMR fingerprinting already at medium ageing state. The figure below will be added in the Supplementary material. The text (lines 198-200 of the first submission) will be changed into:

"In the NMR spectra corresponding to a "medium" SOA oxidation level, the resonance at 0.83 ppm of chemical shift, arising from one of the two gem-methyls of pinonic acid, accounts for only 0.3% of the total integral of the spectrum, while it represented 2 – 3 % in the fresh SOA samples."

[Figure]

*- Section 3.1.2, line 218: The Authors state that "moderately aged SOA show mainly the two singlets of phthalic acid" in the aromatic region. The singlets of phthalic acid should appear/resonate at approximately 8.1 ppm. This assignment is unclear in the spectra of naphthalene SOA (Figure 2) due to the presence of a broad resonance at approximately 7.2-8.3 ppm.*

**REPLY:** Phthalic acid is a diprotic acid, and it can resonate in a range of chemical shifts depending on protonation sate, hence on pH of the aqueous solution. We have performed the NMR analysis of naphthalene SOA in unbuffered solution, therefore we expect the chemical shifts of carboxylic acids to differ to a certain extent between samples. Below, we show a blow-up of Figure 2 of the original manuscript showing the aromatic region in more details. Most of the spectra show a very small singlet at 8.40 – 8.46 ppm which must be attributed to formate (marked with "F" in the figure). The 0.06 ppm difference in chemical shift between samples must be caused by pH differences in the solutions. At the bottom of the figure, we added the spectrum of a phthalic acid standard (potassium hydrogen phthalate) in buffered solution (a potassium deuterated formate/formic acid buffer with pH~3.8, the same used for the isoprene experiments). The position of the formate peak indicates that the pH of the phthalic acid standard solution is approximately the same of sample Nap#2. Our test clearly indicates that the two complex resonances at 7.56 – 7.59 and 7.71-7.74 in sample Nap#2 - and of analogous peaks in samples Nap#4 and Nap#5 (marked with "P" in the figure) - must be attributed to phthalic acid. The other samples (Nap#1, Nap#3 and the blank) exhibit a system of peaks with a fine structure similar to that of the standard of phthalic acid but resonating a lower chemical shifts (between 7.4 and 7.5 ppm) and marked "P' " in the figure. On the basis of the position of the formate peak, approaching 8.46 ppm, in these samples, the peak systems P' are presumably from phthalic acid, while in different pH conditions with respect to samples Nap#2, #4 and #5 and the phthalic acid standard solution.

[Figure]

Below, we present the same figure, with chemical shift range between 7.3 and 7.8 ppm enlarged, showing the fine structure of the phthalate resonances.

[Figure]

- *Section 3.1.2, lines 218-221: Besides exhibiting NMR peaks between 3.5 and 6.0 ppm, all spectra also exhibit noticeable NMR peaks between 1 and 2 ppm. Could these NMR resonances be still a consequence of the presence of colloidal hydrophobic material in solution, or they could be attributed to aliphatic structures derived from the ageing process?*

**REPLY:** The referee is right in pointing out the presence of multiple resonances between 1 and 2 ppm in the spectra of naphthalene SOA. It should be noted, however, that most of them are also found in the blank (spectrum in grey in the Figure 2). The inspection of such peaks in the blank suggests that $\alpha$-pinene oxidation products, such as pinonic acid (peaks marked with "Po" in the figure below) and pinic acid (peaks marked with "Pi") together with other unidentified compounds (marked "X") from previous experiments have caused contaminations of the reactor during the naphthalene experiments (see figure below). Based on these results, it is doubtful that any of the naphthalene SOA samples exhibited genuine resonances in the range between 1 and 2 ppm of chemical shift.

[Figure]

- *Section 3.1.2, lines 224-229: In my opinion, the broad aromatic band between 6.5 and 8.5 ppm is totally indiscernible in the spectrum of the most aged naphthalene SOA sample. I am also compelled to disagree when the Authors conclude that "there is no clear trend in the formation/disappearance of aromatic and aliphatic bands with ageing". In my opinion, the 1H NMR spectrum of the most aged sample clearly indicate the disappearance of resonance in the aromatic region, whereas a few NMR peaks resonate in the aliphatic region (chemical shifts = 3.5 and 5.0 ppm). The Authors should include additional explanations to support their statement/conclusion.*

**REPLY:** The inspection of the spectral feature of sample Nap#1 (the most aged naphthalene SOA sample) is challenging due to the very small loadings and the interference from the reactor blank (see comment above). If we restrict our analysis to chemical shifts higher than 2.5 ppm – to get rid of most blank signals – and enlarge the y scale of the spectra, we can have closer inspection of the resonances in sample Naph#1 (see figure below). In spite of the poor signal-to-noise ratio of this spectrum, a very broad aromatic band between 6.7 and 8.2 ppm is still visible in Nap#1, whereas the baseline is completely flat in the blank spectrum. The aromatic band is actually structured into two sections, with one from phenolic compounds (from 6.7 to 7.2

ppm) and a second one arising from aromatic structures substituted with electron-withdrawing groups (from 7.2 to 8.2 ppm). The latter is relatively much less intense than in the other naphthalene SOA samples. In conclusion, the small sample amount for Nap#1 prevents an accurate quantification of aromatic compounds. The Referee is right in noticing that the aliphatic groups between 3.5 and 5 ppm are much more concentrated in this sample with respect to aromatics; however, it is not entirely true that aromatic groups have completely disappeared in this sample.

[Figure]

The text in the manuscript will be revised and implemented as follows:

"Despite the low naphthalene SOA concentration, a broad aromatic band between 6.5 and 8.5 ppm and the same signals found between 3.5 and 6.0 ppm seen in the samples with a medium O/C ratio are still visible in this most aged naphthalene SOA spectrum. However, the band from oxygenated functional groups between 3 and 4.5 ppm becomes relatively more intense with respect to aromatics compared to SOA samples of smaller ageing state. Compared to $\alpha$-pinene SOA, the [1]H-NMR fingerprint of naphthalene SOA appears less sensitive to variations in the OH exposure between the low and the medium level of exposure. More substantial changes can be found for the most oxidized sample, which are only partly visible due the low signal-to-noise ratio of the spectrum."

*- Section 3.2, lines 275-276: The chromatograms in Figure 5 suggest that the signal intensity of the chromatographic peaks corresponding to mono- and di-acids increases with increasing photochemical age, which seems to contradict the statement "that the TOC mass fraction of mono- and di-acids decreases from 33% to 18%" with increasing photochemical age. Besides, in lines 273-374, the Authors conclude that "a net increase in acidic compounds with photochemical age can be clearly observed". Additional explanations should be provided to clarify these apparent conflicting conclusions.*

**REPLY:** There was a mistake in the text. The TOC mass fraction decreasing from 33% to 18% is only for mono-acids, not for total mono-/di-acids. In fact, the fractional content of di-acids remained fairly stable during ageing in naphthalene SOA. The HPLC classes distributions for the naphthalene SOA samples are reported in Fig. S4 where the concentrations units are derived by the measured TOC content in the eluted chromatographic fractions instead of the integrals of chromatographic peaks. On the other hand, peak area and TOC content are clearly correlated for each HPLC fraction type (see figure below).

[Figure]

The text will be revised as follows:

"However, the HULIS content, initially small, increases substantially and progressively with ageing. With increasing photochemical age, the TOC mass fraction of mono- and di-acids decreases from 33% to 18% and from 34% to 33% respectively, while the fraction of PA/HULIS increases from 11% to 30%".

Below we report Figure 5, reformatted also taking into account Referee 2's suggestions. We believe that the reduction of mono-acids and the relatively stable content of di-acids during ageing is clear from both the TOC measurements and the evolution of chromatographic peaks in Fig. 5.

[Figure]

Figure 5.

Finally, we improved Section 3.2 with more clear references to the TOC data of the HPLC fractions presented in Figures S3 and S4 of the Supplementary. Additional observations on the distribution of HPLC classes is now reported at the end of Section 3.2.

*- Section 3.2, line 282: the abbreviature "NC" for "neutral compounds" should have been previously defined in line 259.*

**REPLY:** It is in fact defined in the Methods section (2.4).

*- Section 4, lines 303-305: In Figure 7, it is unclear which marker correspond to ambient PEGASOS WSOC for the different oxidation levels.*

**REPLY:** The figure reports correlation coefficients. Each value (one point in the figure) represents the Pearson correlation coefficient for one individual SOA spectrum with respect to the reference spectrum for ambient

WSOC (i.e., the same spectrum in Figure S1 upon binning). We have re-formatted the figure (taking into account also the comments by Referee 2):

[Figure]

Figure 7.

*- Section 4, lines 321-322: In the sentence "the H-NMR spectra of alpha-pinene SOA most closely mimic the functional group distributions of the ambient WSOC sample obtained in PEGASOS", are the Authors referring to the spectrum in Figure S1? If so, please redirect the reader to Figure S1.*

**REPLY:** True. We will correct the text.

*- Section 4, lines 331-335: The Authors conclude that the correlation coefficients shown in Figure 7 for the NMR spectra of alpha-pinene vs. ambient WSOC are smaller than those between the HR-ToF-AMS spectra of PAM-generated SOA vs. ambient OOA reported in reference Lambe et al. (2011). Could this difference be explained by the fact that Lambe et al. (2011) is focusing on ambient OOA, which is a fraction of total organic aerosols (OA) and a proxy for SOA, whereas in this study the Authors are comparing the NMR spectra of alpha-pinene with those from the whole WSOC fraction which probably also includes a small fraction of primary OA? Secondly, I did not find Figure 9 in reference Lambe et al. (2011).*

**REPLY:** The Referee is correct, the correlation coefficients between lab-generated SOA and ambient OOA AMS were in fact reported in Fig. 5 in Lambe et al. (2011) not Fig. 9. We have changed the text. We acknowledge that the lack of full overlap between OOA and WSOC in ambient aerosol complicates the interpretation of the different results obtained in this study with respect to Lambe et al. (2011). Regarding the Referee's comment on the overlapping between OOA and WSOC, we have reasons to believe that OOA are not a subclass of WSOC: recent findings reported by Xu et al. (2017) indicate that ambient OOA can be fully or only partially water-soluble depending on their oxidation degree. We will add a sentence in the text to make the reader more aware of the OOA vs. WSOC issue:

"It should be noted, finally, that a comparison of the AMS and NMR techniques with respect to their ability to trace chemical ageing in laboratory SOA and ambient oxidized aerosols is challenged by the incomplete overlap between the classes of organic compounds contributing to OOA and to WSOC (Xu et al., 2017)."

*3. Technical corrections:*

*- Section 2.1, Tables 1 and 2: There is some information missing from the "Oxidation Level" column for Pin#3 (Med.) and Pin#5 (Low) in Table 1, and for Iso#2 (med) in Table 2. Could you please also clarify the meaning of "f44" in these Tables? Is this corresponding to the m/z marker of COOH formation withdrawn from the AMS data? If so, this should be clarified in the manuscript.*

**REPLY:** "f44" refers to the fraction of the m/z 44 signal with respect to total OA in the AMS electron impact MS spectra. We will include a statement of explanation in Section 2.1:

"By varying the concentrations of OH inside the PAM reactor, SOA with different oxidation state could be obtained. For instance, the OH exposure varied from 2.0E11 molec./cm$^3$*s to 2.1E12 molec./cm$^3$*s between the $\alpha$-pinene experiments and the resulting SOA oxidation degree – traced by the "f44" parameter (i.e., the fraction of the m/z 44 signal with respect to the total OA) – increased from 0.05 to 0.24."

We thank the Referee for pointing out missing f44 values from Table 1: Pin#3 oxidation level = med (f44 = 0.11), and Pin#5 oxidation level = low (f44 = 0.05). The AMS was instead not operative during experiment Iso#2 (Table 2).

*- Section 2.2, line 144: Figure S1 shows the 1H NMR spectrum of an ambient WSOC sample collected in San Pietro Capofiume and not the 1H NMR spectra of methanol extracts from isoprene SOA samples. Instead, it could be Figure S2?*

**REPLY:** We apologize for the wrong reference and will correct it in the revised manuscript.

*- Section 2.4, line 167: I did not find the full HPLC-UV-TOC analytical protocol in the Supplementary material.*

**REPLY:** The full description of the analytical protocol can be found in Mancinelli et al. (2007). We will change the sentence "The full HPLC-UV-TOC analytical protocol is reported in the Supplementary material" into "The HPLC column and chromatographic conditions used in this study were the same as in Mancinelli et al. (2007)"

**References:**

Lambe, A. T., Chhabra, P. S., Onasch, T. B., Brune, W. H., Hunter, J. F., Kroll, J. H., Cummings, M. J., Brogan, J. F., Parmar, Y., Worsnop, D. R., Kolb, C. E. and Davidovits, P.: Effect of oxidant concentration, exposure time, and seed particles on secondary organic aerosol chemical composition and yield, Atmos. Chem. Phys., 15, 3063–3075, 2015.

Xu L., Guo H., Weber R. J., Ng N. L.: Chemical characterization of water-soluble organic aerosol in contrasting rural and urban environments in the southeastern United States, Environ. Sci. Technol., 51, 78–88, 2017.

---

## Author Comment (AC2) · 4 Jul 2017

**Replies to Referee 2's comments.**

We thank the Referee for the very through review. Below we copy his/her comments (in italics) and provide point-by-point replies.

**General Comments:**

This manuscript describes the chemical analysis of laboratory-generated secondary organic aerosol (SOA) from isoprene, alpha-pinene, and naphthalene precursors. The authors generate SOA within a Potential Aerosol Mass (PAM) oxidation flow reactor, varying actinic flux to achieve various overall OH exposures, collect SOA on quartz fiber-filters, and then extract water soluble organics for analysis using 1H-NMR and HPLC. The authors find that with increased OH exposure 1H-NMR analysis shows further oxidized organic material being generated, consistent with spectroscopic features of known oxidation products from the studied systems. In comparison to 1H-NMR and HPLC analysis of ambient samples taken from the Po Valley, Italy, and Cabauw, Netherlands, the authors find that highly-oxidized alpha-pinene SOA 1H-NMR spectroscopic features from similarly aged alpha-pinene and naphthalene most resemble that of the ambient samples.

Overall the manuscript is clearly written and the experimental setup and analysis straight forward. The results have potential to be of interest to ACP readers, though I think improvements are needed in discussing and considering the true atmospheric relevance of the current work. In particular, the inclusion of the isoprene results seem superfluous and of confusing atmospheric relevance because of potentially contradictory experimental conditions used and results obtained, and also because the selected ambient samples were not taken from areas notable for isoprene-dominated chemistry. Additional experimental methodology should be clarified as well. After specific and technical comments are addressed below, this manuscript may be suitable for publication in ACP.

**REPLY:** We thank the Referee for his/her through review. We agree on the fact that the correlation analysis between non-IEPOX isoprene SOA spectra and an ambient WSOC spectrum representing a polluted environment, where low-NOx chemistry is mostly suppressed, can be misleading. We have therefore revised Section 4 of the manuscript, by including a new comparison between the isoprene SOA spectral data and those from an ambient WSOC sample from a pristine continental environment. Such spectrum will be reported as a new figure in the Supplementary Information:

Figure S1b.

The WSOC sample (PM1) considered for comparison with the isoprene SOA samples was collected at a pasture site in the Amazon basin (State of Rondônia, Brazil) from 12 to 14 November 2002. The sampling dates correspond to the beginning of the rainy season in this region of the tropics, when isoprene SOA formation from nearby forest emissions takes over local biomass burning sources (Decesari et al., 2006). Full documentation about the organic composition, including NMR data, of the aerosol samples collected during the 2002 field campaign in Brazil can be found in Tagliavini et al. (2005).

The results of the comparison between PAM-generated isoprene SOA samples and the Brazilian WSOC sample are reported in the new Figure 7 of the manuscript:

**Figure 7.**

The first paragraph of Section 4 of the manuscript will be changed as follows:

"In this section, the NMR and HPLC results obtained for the isoprene,  $\alpha$ -pinene and naphthalene SOA systems are compared with ambient OA samples. First, we investigated the similarity between the 1H-NMR spectral profiles of SOA with those "typical" of ambient non-biomass-burning WSOC. For this purpose we used one sample of PM1 collected during the 2012 PEGASOS field campaign (Sandrini et al., 2016) in the rural Po Valley (Italy) which can be considered representative for a continental rural "near-city" site (according to the criteria of Putaud et al., 2010). A second PM1 sample was collected at a rural site in the State of Rondônia (Brazil) during the 2002 SMOCC field campaign, and, more precisely, during the early rainy season, when local biomass burning sources had largely ceased and the organic composition of submicron particles was dominated by biogenic emissions (Decesari et al. 2006). The ambient WSOC and laboratory SOA spectra were binned to 400 points in order to remove the variability in chemical shifts due to, e.g., different pH conditions during the analyses of the samples. Figure 7 shows the correlation between the SOA spectra and the reference spectra of ambient WSOC: the  $\alpha$ -pinene SOA and naphthalene SOA spectra were compared to the Po Valley WSOC sample, while the isoprene SOA spectra were compared to the Amazonian sample. There is good correlation (0.62 < r < 0.92) between the NMR spectra of  $\alpha$ -pinene SOA, at all oxidation levels, with the spectrum of the Po Valley PM1 sample. This finding is in line with modelling results and previous experimental findings indicating that the organic composition in northern Italy in the summertime is dominated by biogenic SOA (Bessagnet et al., 2008; Gilardoni et al., 2011). A moderate positive correlation was also found between the spectra of isoprene SOA and of the PM1 sample from rural Brazil (r = 0.52, 0.54). It should be noted that the relative humidity at this pasture site is variable during the day and very high overnight during the rainy season, based on the meteorological data presented by Betts et al. (2009). Therefore, biogenic aerosols are expected to include also isoprene SOA forming through the IEPOX route (Hu et al., 2015), which is not accounted for by our laboratory experiments. Finally, the naphthalene SOA spectra exhibit zero or negative correlations with the Po Valley WSOC spectrum (-0.15 < r < -0.02). [...]"

We believe that the new version of the discussion section (containing the comparison with the ambient aerosol data representative for an isoprene-dominated environment) supports the atmospheric relevance of the isoprene SOA spectral data presented in this study. It should be noted, finally, that to our knowledge this is the first time that isoprene SOA NMR data - including the spectra of marker compounds (2-methyl-tetrols) and the spectral fingerprints for complex secondary organic compound mixtures generated in laboratory from isoprene photo-oxidation - are presented in the literature. We also modified Section 3.1.3 to highlight the implications of our NMR measurements of the isoprene SOA samples:

"The isoprene SOA 1H-NMR spectra profiles were all very similar (an example is provided in Figure 3). The comparison with literature data (Budisulistiorini et al. 2015) led to the unambiguous identification of 2-methyltetrols, clearly responsible for the two singlets at 1.12 ppm (methylic H atoms of methylerythritol) and 1.13 ppm (methylic H atoms of methylthreitol) and for a series of multiplets between 3.4 and 3.9 ppm. Methylerythritol is more abundant (60% of the sum of the two, as an average between two samples extracted in water and three extracted in methanol) than methylthreitol. The spectra show the occurrence of only two diastereomers among the possible four ones (González et al. 2011), indicating that the formation of methyltetrols is stereoselective, as already proposed by Cash et al. (2016) on the basis of a theoretical analysis of the IEPOX chemistry, and in contrast with the conclusions of González et al. (2011) claiming that methyltetrols are produced in laboratory conditions only in racemic mixtures."

More implications of the NMR analysis of isoprene SOA will be presented in response to the Referee's 6th specific comment.

Specific comments:

1. Lines 31-32: Please clarify the definition of the maximum yield %'s listed. Are these yields mass of HULIS generated per mass of precursor reacted away, similar to an SOA yield? If so, please provide quantification/calibration information. Rather, are they fraction of WSOC that is designated as HULIS (as presented in SI Figs. S3, S4)?

**REPLY:** The yields refer to the fraction of WSOC (= organic carbon in HULIS with respect to the sum of neutral compounds and all acidic fractions). The sentence will be rephrased as follows:

"Over multiple days of equivalent OH exposure (30 atmospheric days), the formation of HULIS is observed in both  $\alpha$ -pinene SOA and in naphthalene SOA (maximum yields: 16% and 30%, respectively, of total analyzed WSOC), providing evidence of the formation of humic-like polycarboxylic acids in unseeded SOA."

2. Lines 50-79: In light of the discussion here, the authors should provide additional context about the major SOA formation processes governing their particular experiment. Since the experiments are unseeded, what is the relative importance and timescale of nucleation, heterogeneous oxidation, and gas-particle partitioning in generating the characterized SOA? And how do these processes and timescales differ with ambient timescales even if similar total OH exposure is achieved?

The following text will be added to the end of Section 2.1 in the revised manuscript:

"To compare SOA formation processes occurring in oxidation flow reactors and in the atmosphere, two primary assumptions are required. First, we assume the kinetics of laboratory processes occurring at higher oxidant concentrations and shorter exposure times can be extrapolated to atmospheric processes occurring at lower oxidant concentrations and longer residence times. Second, we assume that the extent of nucleation or phase partitioning of SOA is not limited by the shorter residence time in flow reactors. The first assumption is supported by Renbaum and Smith (2011), Bahreini et al. (2012), and Lambe et al. (2015). The second assumption may introduce uncertainty depending on the particle surface area available to promote condensation and the mass accommodation coefficient of the oxidized vapors (Lambe et al., 2015; Shantanu et al., 2017).

3. Line 73: Consider including these additional references in discussing vapor-phase wall losses:

(Matsunaga and Ziemann, 2010; Ye et al., 2016)

**REPLY:** Accepted. We thank the Referee for the useful references.

4. Lines 87-89: This statement does not seem fair or accurately written considering the authors cite work of Cavalli et al., 2006 later for comparing alpha-pinene SOA and plenty of 1H-NMR spectroscopy has been performed on SOA samples from urban areas e.g. (Cleveland et al., 2012). Do the authors want to reword the sentence to specify something specific about novelty of applying the technique to laboratory-generated SOA from known biogenic and anthropogenic SOA precursors?

**REPLY:** We agree. The sentence "The present study is the first application of 1H-NMR spectroscopy to SOA samples produced from the OH oxidation of biogenic and anthropogenic SOA" will be changed into "The present study is among the first applications of 1H-NMR spectroscopy to SOA samples produced from the OH oxidation of biogenic and anthropogenic SOA in the laboratory". We included citations from relevant literature on laboratory experiments of SOA generation, while essentially ignoring the literature on NMR

aerosol field studies (except from a few papers which were cited here for reporting information on methodological aspects).

**5. Lines 151-153: Can the authors clarify if this choice in using the buffer has to do with the specific products expected from non-IEPOX isoprene chemistry that may be subject to chemical shift of the hydrogen atoms?**

**REPLY:** Not specifically for isoprene SOA samples. This is because the isoprene SOA samples were analyzed at a later time with respect to the alpha-pinene and naphthalene SOA ones, and in the meanwhile we realized that including a pH buffer led to a higher stability of chemical shifts.

6. Interpretation of isoprene results in Section 3.1.3 NMR fingerprints of non-IEPOX isoprene SOA: In light of the experimental conditions being dry (according to lines 117) and unseeded, why do the authors attribute the spectroscopic features to 2-methyl tetrols as products in their experiments? These are typically observed as particle-phase products resulting from IEPOX route of SOA formation from isoprene, but the authors claim their conditions may resemble more of non-IEPOX route of SOA formation according to Krechmer et al., 2015 in lines 107-112 and Liu et al., 2016 in lines 249-250. Do the authors suggest that peroxide-equivalents of 2-methyl tetrols (formed via nucleation under their experimental conditions?) would have the exact same NMR fingerprint as 2-methyl tetrols? If so, this really weakens the use of NMR fingerprints with true molecular-level mechanistic understanding. The authors should reconcile their measurements and the mechanism more clearly.

**REPLY:** The two major compounds in our isoprene SOA samples are undoubtedly 2-methyl-tetrols. The NMR resonances of these species are completely consistent with the spectra reported in Fig. S13 of the paper by Budisulistiorini et al. (2015) and with the chemical shift data reported for pure standards of 2-methyl-tetrols by González et al. (2011). On the basis of NMR spectra simulations performed using ACD/Labs software (Advanced Chemistry Development, Inc., 1D NMR Manager & NMR Predictor Suite v.12), the peroxide analogues of methyltetrols must exhibit some resonances above 4 ppm of chemical shift, because hydroperoxyl groups are more electron-withdrawing than hydroxyl substituents. Most importantly, hydroperoxyl substituents can occur in variable number and at diverse positions in the molecules (Riva et al., 2016), resulting in rather complex NMR spectra, whereas our samples show clearly the occurrence of just two individual compounds, which we assign to the two major diastereomers of methyl-tetrols (2-methylthreitol and 2-methyl-erythritol). The reason why our samples of non-IEPOX SOA show prevalently 2-methyltetrols rather than their peroxide analogues, in contrast with the results of Krechmer et al. (2015) and of Riva et al. (2016), is unclear. We can speculate that peroxides self-reactions lead to the formation of alcohols  $(ROOH+ROOH => 2 ROH + H_2O_2)$  in the sample or on the filter surface. Considering that the discovery of non-IEPOX SOA is rather new and little is known about the stability of isoprene peroxides in aerosol samples, we can just acknowledge that more research must be carried out to clarify the discrepancy between our findings and the literature studies. We have included this specific issue in Section 3.1.3:

"The results of Liu et al. (2016), indicating that non-IEPOX isoprene SOA include peroxide-equivalents of methyltetrols, are in agreement with these findings. However, Riva et al. (2016) reported only peroxides for non-IEPOX SOA in unseeded experiments and no methyl-tetrols. It is possible that peroxides decomposed to tetrols in our filter samples during collection downstream the PAM or afterwards during storing. The actual stability of isoprene hydroperoxides in the aerosol itself is largely uncertain, therefore the discrepancy between the findings presented in this study and the results of Riva et al. (2016) cannot be clarified at this stage. In addition, neither Liu et al. (2016) and Riva et al. (2016) reported the presence of carboxylic or keto groups, while our data clearly indicate that these (and/or other acyl groups) are found in the unresolved mixtures of non-IEPOX isoprene SOA, and are responsible for the signal band between 2.0 and 2.6 ppm. Still,

this band is much less intense than that of alcoxyls, which is opposite to what observed for  $\alpha$ -pinene SOA where acyls are by far the main oxygenated aliphatic functional group. Thus, 1H-NMR spectroscopy provides distinct fingerprint for isoprene and monoterpene SOA."

**7. Lines 279-281, Lines 355-361: The authors should clarify if water extraction methods of the organic material from filters can lead to aqueous processing and artificial formation of HULIS.**

**REPLY:** We did not perform the water extraction in an inert atmosphere (e.g., under nitrogen flux), therefore, in principle, we cannot exclude the degradation of labile organic structures. These, however, must be extremely labile species and able to react with dissolved oxygen. The OH concentrations, instead, will be unrealistically too small in the sealed vials where the extraction was performed (the OH molecules in the few cubic centimeters of head space will be just some millions, while we can reach 1 µmol of organic carbon dissolved in the extract). We believe that the main effect of water extraction is the hydrolysis of anhydrides and acetals. It is largely recognized that several of the known types of oligomerization reactions deemed to occur in atmospheric organic particles are in fact reversible (Kroll and Seinfeld 2008). Such oligomers are not expected to survive the water extraction step of our analytical protocols, therefore, the HULIS determined in this study must be considered stable against hydrolysis. We will update the first paragraph of Section 3.2 to specify that:

"The HPLC analysis of fresh  $\alpha$ -pinene SOA extracts shows the presence of compounds unretained by ionexchange columns (neutral compounds) or weakly retained (mono- and di-acids) with a small contribution from compounds having a high retention factor (polyacids, PA, or HULIS), in agreement with previous results obtained from  $\alpha$ -pinene SOA samples generated in environmental chamber experiments (unpublished data). It should be noted, however, that the chromatographic analysis of SOA compounds in water extracts generally does not allow to recover high-molecular weight organic oligomers susceptible to hydrolysis reactions (e.g., polyacetals, Kroll and Seinfeld 2008). The HULIS determined by our method are essentially only the non-hydrolyzable ones, stable in aqueous solutions."

**8. Line 298: The experimental setup is dry. How can the authors be certain that they are generating the same compounds in SOA formation/mimicking atmospherically relevant mechanisms from their chosen precursors and then comparing to WSOC from atmospheric samples?**

**REPLY:** We do not claim that our laboratory study is comprehensive with respect to simulating the full range of possible atmospheric conditions conditioning SOA formation. We just specified that the SOA chemical composition data presented in this study are more representative for the low relative humidity (RH) conditions. This is often the case, for instance, of Mediterranean climates in summertime (the average RH corresponding to the Po Valley sample in Fig. S1 was 35%), but it is also common for continental sites in temperate climates during a high-pressure anomaly, like in the case of the May 2008 field campaign in Cabauw (RH ~ 30% to 50% in the first 2 km of atmosphere, according to Derksen et al., 2011). The other ambient sample considered in this study is the Amazonian PM1 sample from Rondônia (Brazil) introduced for comparison with the isoprene SOA sample (see above). We could not find meteorological data for the collection time of this last sample, therefore we must refer to climatological data for Rondônia, presented by Betts et al. (2009). Note that the sampling location was a deforested pasture site, where water vapor mixing ratio is normally lower than at forest sites in the same region. Our sample was collected at the very beginning of the rainy season, just after the end of the transition period (i.e., the transition between the dry and the wet season). On the basis of the temperature and water mixing ratio data presented by Betts et al. (2009), we can calculate a maximum relative humidity (RH) of 90 - 92% (night-time and early morning hours) and a minimum RH of 65 – 77% in afternoon hours, when the atmospheric mixing layer reaches its maximum

thickness. We can conclude that, during the collection time of the PM1 WSOC sample considered in this study (12 - 14 Nov 2002), the aerosol particles were fully deliquesced overnight, while the liquid water content in daytime was quite variable and certainly not always high, considered the weak hygroscopicity of the particles measured at the pasture site (hygroscopic growth factor < 1.1 at 80% RH, according to Rissler et al., 2006). Therefore, although an IEPOX route of formation of isoprene SOA is widely recognized as a very significant source of organic aerosols in the Amazon basin (Hu et al., 2015), it is not clear whether this is really prevalent respect to non-IEPOX SOA formed at lower RH values at the deforested pasture site considered here.

9. Lines 346-349: While this may be true, Figure 8 seems to suggest that even just fresh naphthalene SOA has the closest WSOC breakdown of HPLC fractions with the ambient OA sample. This suggests that the experimental results are getting the right HPLC fractions, but not necessarily for the right composition/same mechanisms and for that matter the same oxidative aging. Considering the ambient sample is from a rural site, it also seems odd that the naphthalene WSOC breakdown is so close to that observed.

**REPLY:** The Referee is right. However, as we noted in the same section, "naphthalene and other polyaromatic hydrocarbons are co-emitted with many other anthropogenic IVOCs and VOCs including aliphatic compounds in the real atmosphere". We can speculate that aliphatic IVOCs have much lower HULIS formation yields than aromatic ones, therefore the HULIS fraction in fresh anthropogenic emissions must be smaller than the 11% observed for pure naphthalene. The same is valid for anthropogenic SOA in aged plumes: the overall HULIS fraction would be much smaller than the 30% found for pure naphthalene, and becoming more comparable with the 12% observed in the field at a rural site. This is just a hypothesis, but we would like to stress the fact that the HULIS yield of aromatic compounds like naphthalene is probably greater than for other IVOCs (as aromatic compounds undergo oxidative carbon-carbon bond cleavage with much smaller molecular fragmentation than aliphatics) and that consequently the HULIS fraction of both fresh and aged naphthalene SOA must be higher compared to a more complex mixture of IVOCs.

The text in the paper will be revised as follows: "The  $\alpha$ -pinene SOA generated in the PAM reactor at high photochemical age and the fresh naphthalene SOA are characterized by a HULIS amount similar to that of Cabauw samples, while the polyacidic content of aged naphthalene SOA is higher than in the ambient samples. In the real atmosphere, naphthalene is co-emitted with many other reactive VOC and IVOC with potentially very diverse HULIS formation yields, therefore the results presented in Figure 8 do not necessarily mean that the chemical composition of ambient OA in Cabauw is better described by the monoterpene chemistry rather than by anthropogenic IVOC oxidation. On the other hand, these results demonstrate that laboratory experiments of SOA formation can generate complex mixtures of products with the same chromatographic properties of HULIS provided a sufficient extent of photochemical aging using the PAM reactor or related techniques".

Finally, we omitted the pie chart of isoprene SOA from Figure 8, because it cannot be considered representative for the HPLC composition of a polluted site (Cabauw) (see reply to the Referee's first major comment).

10. Please provide additional context for the ambient samples from Cabauw and Po Valley. Are the campaigns noted to be regions of high monoterpene emissions and anthropogenic (aromatic) emissions? They are both described as be rural sites, yet see comment 9 above.

**REPLY:** According to the definition of "sites' catchments" by Henne et al. (2010) ("area in which surface fluxes are expected to create a detectable and significant signal at the receptor sites"), the Cabauw site is affected by aerosol sources from a vast region in north-west Europe, while the station of Ispra in the Po Valley receives

contributions from a more confined area in northern Italy and the great Alpine region. In both areas, the most common land types are managed agricultural land and deciduous forests, the latter being active sources of monoterpenes. In addition, both stations were clustered among the most impacted ones from urban emissions, which include emissions of intermediate volatility aromatic hydrocarbons. We can therefore conclude that monoterpenes and anthropogenic VOCs are expected to impact ambient SOA both in the rural Netherlands (Cabauw) and in the Po Valley (S. Pietro Capofiume, not considered in Henne et al. (2010) is similar to Ispra, though more rural).

Information on sources of SOA in the Po Valley and at the rural site of Brazil has already been included into the revised paragraph of Section 4 upgraded in response of the Referee's first comment. Here we report the new text to integrate into the 3rd paragraph of the same section for introducing the ambient samples from Cabauw:

"The distribution of neutral vs. acidic classes of compounds in ambient WSOC refers to the average of the samples collected at the rural background station of Cabauw in the Netherlands (Paglione et al. 2014b). The station is located downwind from anthropogenic sources and biogenic emissions (terpenes from deciduous forests) over a large sector of north-west Europe (Henne et al., 2010). The HULIS contribution in these samples varied between 15 and 20%, [...]."

11. Figures 1-3: Similar to the regions pointed out in HPLC results for neutral, mono/di-acids, HULIS, it would be helpful if there were similar labels pointing out characteristic bond features associated with the chemical shifts in the 1H-NMR spectra.

**REPLY:** Below are the new versions of Figures 1, 2 and 3 with highlighted the spectral regions characteristic of functional groups.